# Transformer for Partial Differential Equations' Operator Learning

**Zijie Li**                                                                *zijieli@andrew.cmu.edu*
*Department of Mechanical Engineering*
*Carnegie Mellon University*

**Kazem Meidani**                                                       *mmeidani@andrew.cmu.edu*
*Department of Mechanical Engineering*
*Carnegie Mellon University*

**Amir Barati Farimani**                                                      *barati@cmu.edu*
*Department of Mechanical Engineering*
*Carnegie Mellon University*

**Reviewed on OpenReview:** *https://openreview.net/forum?id=EPPqt3uERT*

## Abstract

Data-driven learning of partial differential equations' solution operators has recently emerged as a promising paradigm for approximating the underlying solutions. The solution operators are usually parameterized by deep learning models that are built upon problem-specific inductive biases. An example is a convolutional or a graph neural network that exploits the local grid structure where functions' values are sampled. The attention mechanism, on the other hand, provides a flexible way to implicitly exploit the patterns within inputs, and furthermore, relationship between arbitrary query locations and inputs. In this work, we present an attention-based framework for data-driven operator learning, which we term Operator Transformer (OFormer). Our framework is built upon self-attention, cross-attention, and a set of point-wise multilayer perceptrons (MLPs), and thus it makes few assumptions on the sampling pattern of the input function or query locations. We show that the proposed framework is competitive on standard PDE benchmark problems and can flexibly be adapted to different types of grids. [1]

## 1 Introduction

Many of the phenomena in different fields of science and engineering are modeled by Partial Differential Equations (PDEs). Formulating and solving the governing PDEs help us understand, predict, and control complex systems in physics, engineering, etc. Various numerical schemes approach the problem of solving the PDEs via discretizing the solution space and solving a finite-dimensional problem. However, solving complex spatiotemporal PDEs demands a high-resolution discretization in both time and space which is computationally expensive.

A group of data-driven PDE solvers aim to directly learn the solution from the data observations without any prior knowledge of the underlying PDE. These methods usually rely on a supervised deep learning architecture which imposes an inductive bias that can suit the problem in hand. The use of convolutional layers for structured grids (Khoo et al., 2021; Zhu & Zabaras, 2018; Bhatnagar et al., 2019; Pant et al., 2021), and graph layers for learning unstructured local relations in a system (de Avila Belbute-Peres et al., 2020; Iakovlev et al., 2021; Li & Farimani, 2022; Li et al., 2022; Ogoke et al., 2021) are just two instances of these

---

[1]Code is available at: https://github.com/BaratiLab/OFormer.

models. Learning such function mappings, however, is restricted to the specific resolution and the geometry observed in the training phase.

The pioneering work DeepONet (Lu et al., 2021a) introduces a new data-driven modeling paradigm that aims to learn the solution operator of PDEs and provides a practical realization of the general universal nonlinear operator approximation theorem (Chen & Chen, 1995). Concurrent work Graph Neural Operator (Anandkumar et al., 2020) proposes another operator learning architecture that leverages learnable iterative kernel integration. The promising features of data-driven operator learning, in particular the generalization capability within a family of PDE and the potential adaptivity to different discretizations, have given rise to a wide array of recent research works (Li et al., 2020b; 2021a; Wen et al., 2022; Rahman et al., 2022; Cao, 2021; Kissas et al., 2022; Lu et al., 2022; Cai et al., 2021; Bhattacharya et al., 2021; Nelsen & Stuart, 2021; Gupta et al., 2021). The training of data-driven operator can also be augmented with physics prior to regularize the search space of optimal model, bridging the physics-informed machine learning (Raissi et al., 2019; Sun et al., 2020) with data-driven operator learning (Wang et al., 2021a; Li et al., 2021b; Wang et al., 2021b).

In practice, the data-driven operators are trained based on the finite-dimensional approximation of the input and output functions, where functions are sampled on some discretization grids. Many of the learned operators, despite showing promising performance on solving PDEs, are limited to have the same discretizations for the input and output (Anandkumar et al., 2020; Li et al., 2020b), fixed input grids (Lu et al., 2021a) or to have equi-spaced grid points (Li et al., 2021a; Gupta et al., 2021; Cao, 2021). As stated in Cao (2021), attention (Vaswani et al., 2017) can be viewed as certain types of learnable integral operators and its inference is equivalent to calculating the integral using input points as the quadrature points, which poses few constraints on the underlying discretization grid. Leveraging this property, we propose a flexible attention-based operator learning framework. Concretely, our main contribution includes: (i) A cross-attention module that enables discretization-invariant query of output function; (ii) A latent time-marching scheme that reduces time-dependent PDE into latent ODE (ordinary differential equation) with a fixed time interval; (iii) A flexible Transformer architecture that can take a varying number of input points and can handle both uniform and non-uniform discretization grids.

## 2 Related works

**Learned PDE Solvers**   In the past decade, there have been a number of studies on applying deep learning architectures for solving PDEs (Lu et al., 2021b; Karniadakis et al., 2021; Brunton et al., 2020). Physics-informed methods (Raissi et al., 2019; Sun et al., 2020; Sirignano & Spiliopoulos, 2018; E & Yu, 2018) exploit the PDE supervision to approximate the solution (usually parameterized by a neural network) in an aggregated space-time domain. They are accurate and generally mesh-agnostic, yet require retraining across different instances of a PDE (e.g. change of coefficient). Without knowledge of the PDE, the solutions can also be learned purely from the data. A common strategy for such task is to first spatially encode the data and then use various schemes to evolve in time. Convolutional layers (Stachenfeld et al., 2022; Wang et al., 2020; Bhatnagar et al., 2019; Kochkov et al., 2021), kernels (Saha & Mukhopadhyay, 2021), and graph-based layers (de Avila Belbute-Peres et al., 2020; Iakovlev et al., 2021; Li & Farimani, 2022; Li et al., 2022) are some of the common architectures for learning the spatial relations in a PDE. In this study, we show that transformers can be pliable yet effective spatial encoders.

**Time-marching in the latent space**   Learning the temporal component of the spatio-temporal PDEs can be challenging. A common and straightforward approach is to follow an Encoder-Process-Decoder (EPD) scheme to map the input solution at time $t$ to the solution at next time step (Brandstetter et al., 2022; Stachenfeld et al., 2022; Pfaff et al., 2020; Sanchez-Gonzalez et al., 2020; Pant et al., 2021; Hsieh et al., 2019). An alternative approach which can significantly reduce the computational complexity and memory usage is to propagate the dynamics in the latent space (Lee & Carlberg, 2021; Wiewel et al., 2019). Therefore, once the encoding from observation space to the latent space is done, the system can evolve in time using Recurrent Neural Networks (RNNs) such as LSTM (Wiewel et al., 2019) or even linear propagators based on the assumptions of learning Koopman operator (Morton et al., 2018; Li et al., 2020a; Lusch et al., 2018; Takeishi et al., 2017; Pan & Duraisamy, 2020). In this work, we design a latent time-marching architecture

that brings about scalability of the model's attention encoder and allows the model to unroll for the future time steps during training.

**Neural Operators**  The pioneering work DeepONet (Lu et al., 2021a) introduces the first practical realization of universal operator approximation theorem (Chen & Chen, 1995), which can further be integrated with prior knowledge of the system (Wang et al., 2021a;b). In another group of works, the infinite-dimensional solution operators are approximated by iterative learnable integral kernels Anandkumar et al. (2020). Such kernel can be parameterized by message-passing (Gilmer et al., 2017) neural networks (Anandkumar et al., 2020; Li et al., 2020b) or convolution in the Fourier domain (Li et al., 2021a). Kovachki et al. (2021) further provide a theoretical analysis of the approximation capacity of kernel-integral-based learning framework. Fourier Neural Operator (FNO) (Li et al., 2021a) and its variants (Li et al., 2021b; Guibas et al., 2022; Tran et al., 2021) have shown promising results in various applications (Pathak et al., 2022; Wen et al., 2022). These integral kernels can also be learned in the multiwavelet domain (Gupta et al., 2021; Tripura & Chakraborty, 2022). Furthermore, other than Fourier bases or multiwavelet bases, the bases needed for approximating the solution operator can be learned and composed via attention (Cao, 2021).

**Transformer for physical system**  After the groundbreaking success in natural language processing (Vaswani et al., 2017; Devlin et al., 2018), attention has also been demonstrated as a promising tool for various other machine learning tasks (Dosovitskiy et al., 2021; Tunyasuvunakool et al., 2021; Veličković et al., 2017). Following the success in these fields, several previous works (Geneva & Zabaras, 2022; Han et al., 2022; Shao et al., 2022; Cao, 2021; Kissas et al., 2022) have explored using attention to model and simulate physical system, which can generally be divided into two orthogonal lines. In the first group, attention layers are used to capture the structures and patterns lie in the PDEs' spatial domain (Cao, 2021; Kissas et al., 2022; Shao et al., 2022). On the contrary, in the second group, the attention is used to model the temporal evolution of the system while the spatial encoding is done by other mechanisms like CNNs or GNNs (Geneva & Zabaras, 2022; Han et al., 2022). Our model lies in the former group where attention-based layers are used for encoding the spatial information of the input and query points, while the time marching is performed in the latent space using recurrent MLPs.

## 3  Method

### 3.1  Attention mechanism

**Standard attention**  The standard attention (Vaswani et al., 2017; Bahdanau et al., 2014; Graves et al., 2014; Luong et al., 2015) is an operation defined upon three sets of vectors, namely the query vectors $\{\mathbf{q}_j\}$, key vectors $\{\mathbf{k}_j\}$, and value vectors $\{\mathbf{v}_j\}$. Concretely, for each query $\mathbf{q}_i$, the corresponding output $\mathbf{z}_i$ is calculated as a weighted sum (assuming number of vectors in each set is $n$):

$$\mathbf{z}_i = \sum_{j=1}^{n} \alpha_{ij}\mathbf{v}_j, \quad \alpha_{ij} = \frac{\exp\left(h(\mathbf{q}_i, \mathbf{k}_j)\right)}{\sum_{s=1}^{n} \exp\left(h(\mathbf{q}_i, \mathbf{k}_s)\right)}. \tag{1}$$

The most common choice for the weight function $h(\cdot)$ is the scaled dot-product (Vaswani et al., 2017): $h(\mathbf{q}_i, \mathbf{k}_j) = (\mathbf{q}_i \cdot \mathbf{k}_j)/\sqrt{d}$, with $d$ being the channel of vectors $\mathbf{q}_i$ and $\mathbf{k}_j$. The scaled dot-product can be written in matrix representation as: $\mathbf{Z} = \text{softmax}\left(\mathbf{Q}\mathbf{K}^T/\sqrt{d}\right)\mathbf{V}$, where $\mathbf{z}_i, \mathbf{q}_i, \mathbf{k}_i, \mathbf{v}_i$ are the $i$-th row vectors of matrices $\mathbf{Z}, \mathbf{Q}, \mathbf{K}, \mathbf{V}$ respectively. Due to the presence of the softmax, the multiplication $\mathbf{Q}\mathbf{K}^T$ must be evaluated explicitly, resulting in overall complexity of $O(n^2 d)$, which is prohibitively expensive when applying to long sequences.

**Attention without softmax**  In natural language processing and computer vision tasks, the $i$-th row vector of query/key/value matrices, $\mathbf{q}_i/\mathbf{k}_i/\mathbf{v}_i$, are usually viewed as the latent embedding of a token (e.g. a word or an image patch). Other than the "row-wise" interpretation of attention, Cao (2021) proposes that each column in the query/key/value matrices, can potentially be interpreted as the evaluation of a learned basis function at each point. For instance, $\mathbf{V}_{ij}$ (also the $i$-th element of the $j$-th column vector: $(\mathbf{v}^j)_i$) can be viewed as the evaluation of the $j$-th basis function on the $i$-th grid point $x_i$, i.e. $\mathbf{V}_{ij} = v_j(x_i)$, and similarly

for the matrices $\mathbf{Q}, \mathbf{K}$ (with each column represent the sampling of basis function $q_j(\cdot), k_j(\cdot)$). Facilitated by the learnable bases interpretation, Cao (2021) proposes two types of attention that can be viewed as the numerical quadrature of different forms of integrals:

$$\text{Fourier type:} \quad (\mathbf{z}_i)_j = \frac{1}{n} \sum_{s=1}^n (\mathbf{q}_i \cdot \mathbf{k}_s)(\mathbf{v}^j)_s \approx \int_\Omega \kappa\left(x_i, \xi\right) v_j(\xi) d\xi, \tag{2}$$

$$\text{Galerkin type:} \quad (\mathbf{z}^j)_i = \sum_{l=1}^d \frac{(\mathbf{k}^l \cdot \mathbf{v}^j)}{n}(\mathbf{q}^l)_i \approx \sum_{l=1}^d \left(\int_\Omega (k_l(\xi)v_j(\xi))d\xi\right) q_l(x_i), \tag{3}$$

where $\Omega$ denotes the spatial domain that is discretized by $n$ points $\{x_i\}_{i=1}^n$, and the dot product $\mathbf{q}_i \cdot \mathbf{k}_s$ in equation 2 approximates the learnable kernel function $\kappa\left(\cdot, \cdot\right)$ at $(x_i, x_j)$. The approximation capability of Fourier type attention and Galerkin type attention are studied theoretically at (Kovachki et al., 2021; Choromanski et al., 2020) and (Cao, 2021) respectively. Inspired by the Gram-Schmidt process, Cao (2021) heuristically chooses layer normalization (Ba et al., 2016) to normalize $\mathbf{q}_i, \mathbf{k}_s$ in equation 2 and $\mathbf{k}^l, \mathbf{v}^j$ in equation 3. But if we further exploit the basis function property of these column vectors, it is also reasonable to impose non-learnable instance normalization (Ulyanov et al., 2016) on each column vector such that their $L_2$ norm equals to one, i.e. $\left\|\mathbf{v}^j\right\|_2 = 1$. In practice, we find this slightly improves the model's performance, and thus we adopt instance normalization throughout the model. Equipped with the normalization schemes, equation 2 and equation 3 in matrix multiplication form write as:

$$\text{Fourier type:} \quad \mathbf{Z} = \frac{1}{n}\widehat{\mathbf{Q}}\widehat{\mathbf{K}}^T \mathbf{V}; \quad \text{Galerkin type:} \quad \mathbf{Z} = \frac{1}{n}\mathbf{Q}(\widehat{\mathbf{K}}^T \widehat{\mathbf{V}}), \tag{4}$$

where $\widehat{\cdot}$ denotes a column-wise normalized (via instance normalization) matrix. The above integral-based attention mechanisms serve as simple but powerful building blocks for PDE operator learning model. The softmax-free property allows both Fourier type and Galerkin type to be computed efficiently in the form (omitting the normalization scheme): $\mathbf{Z} = \mathbf{Q}(\mathbf{K}^T \mathbf{V})/n$, since matrix multiplication is associative.

**Cross-attention** While the aforementioned attention mechanisms are flexible with respect to the discretization of input domain, the fact that $\mathbf{Q}, \mathbf{K}, \mathbf{V}$ are essentially different linear projections of the same input feature embedding $\mathbf{H}$ makes the input $\mathbf{H}$ and output $\mathbf{Z} = \mathbf{Q}(\mathbf{K}^T \mathbf{V})/n$ locked to the same fixed grid $\{x_i\}_{i=1}^n$. To decouple the output domain from the input domain and allow for arbitrary query locations, we extend the above attention mechanisms from self-attention to cross-attention, where the query matrix $\mathbf{Q}$ is encoded from a different input. To be more specific, the $i$-th row vector $\mathbf{q}_i$ in the query matrix $\mathbf{Q}$ is the latent encoding of query point $y_i$, where $\{y_i\}_{i=1}^m$ denotes the discretization of the output domain and is not necessarily same as the input grid points $\{x_i\}_{i=1}^n$. Hence, leveraging the cross-attention mechanism, we can query at arbitrary locations which are independent of the input grid points. If we adopt the learned bases interpretation from previous section, the cross-attention mechanism can also be viewed as a weighted sum of three sets of basis functions: $\{k_l(\cdot), v_l(\cdot), q_l(\cdot)\}_{l=1}^d$ with $k_l(\cdot), v_l(\cdot)$ defined on the input discretization grid $D(x) : \{x_i\}_{i=1}^n$ and $q_l(\cdot)$ defined on the query discretization grid $D(y) : \{y_i\}_{i=1}^m$, which serves as the finite-dimensional approximation space's basis for the final target function:

$$z_s(y_j) = \sum_{l=1}^d \frac{\sum_{i=1}^n k_l(x_i)v_s(x_i)}{n} q_l(y_j). \tag{5}$$

If we drop the point-wise feed forward network (FFN) after the attention layer, which is a MLP with two layers (omitting the bias term): $\mathbf{x} \leftarrow \sigma\left(\mathbf{x}\mathbf{W}^{(1)}\right)\mathbf{W}^{(2)}$, where $\mathbf{W}^{(1)}, \mathbf{W}^{(2)} \in \mathbb{R}^{d \times d}$ are learnable weight matrices and $\sigma(\cdot)$ is a non-linear activation function, and then directly make $z_s(y_j)$ as output, the cross attention in the equation 5 is a special case of the universal operator approximator proposed in DeepONet (Lu et al., 2021a) with $\sum_{i=1}^n k_l(x_i)v_s(x_i)/n$ as the output of branch net and $q_l(y_j)$ as the output of trunk net. But in practice, we find that appending FFN after attention improves the model's performance and a propagating MLP is necessary for time-dependent PDEs (see Figure 2). Another notable difference is that the number of parameters in the attention layer does not depend on the input's discretization and thus it is applicable to samples with varying grids without re-training. More broadly speaking, the query matrix $\mathbf{Q}$, whose column space is spanned by the learnable query basis functions $\{q_l(\cdot)\}_{l=1}^d$, loosely resembles the induced points in Set Transformer (Lee et al., 2018) and the latent array in Perceiver (Jaegle et al., 2021).

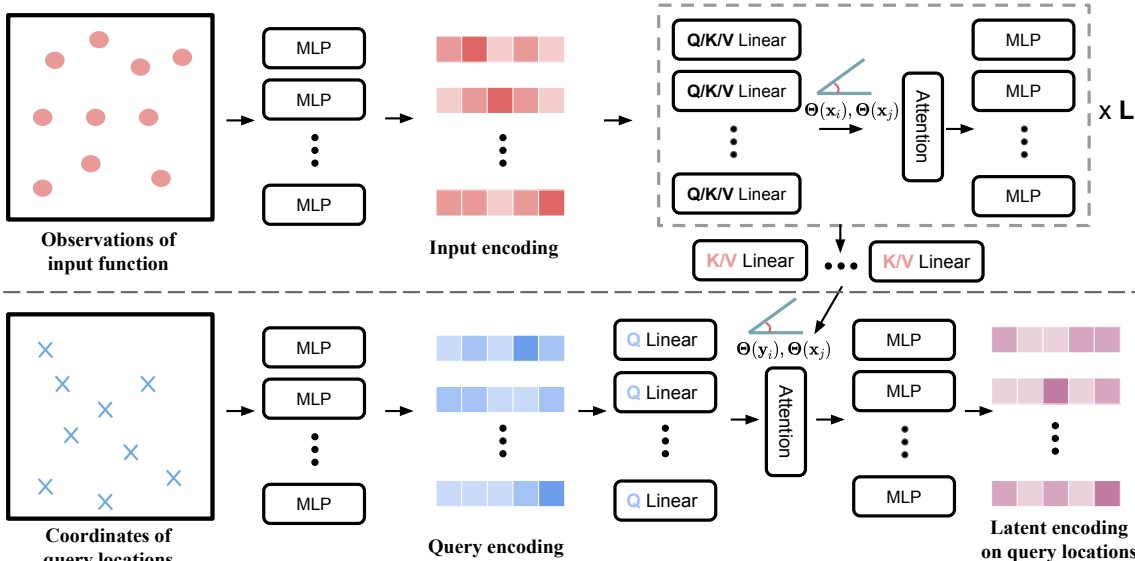

Figure 1: Attention-based encoder architecture. **Top row**: Input encoder that encodes input function information based on locations ($\{x_i\}_{i=1}^n$) where input function is sampled and the sampled function value. **Bottom row**: Query encoder that encodes the coordinates of query locations ($\{y_i\}_{i=1}^m$) and uses encoded coordinates to aggregate information from input encoding via cross-attention. Rotary positional encodings $\boldsymbol{\Theta}(\cdot)$(equation 9) are used in both self-attention and cross-attention.

## 3.2  Model architecture

The proposed model has an Encoder-Decoder architecture that is similar to the original Transformer (Vaswani et al., 2017), where the input is first processed by several self-attention blocks and then attended to the output. However, we only use cross-attention to derive the latent embeddings sampled on query locations and then use recurrent MLPs to propagate dynamics instead of masked attention as in typical Transformer decoder.

**Encoder**  The encoder comprises three components (Figure 1), an input encoder $\phi^{\mathcal{X}}(\cdot)$ that takes the input function's sampling $a(x_i)$ and coordinates of sampling positions $\{x_i\}_{i=1}^n$ as input features, a query encoder $\phi^{\mathcal{Y}}(\cdot)$ that takes the coordinates of query locations $\{y_i\}_{i=1}^m$ as input features, and a cross-attention module to pass system information from input locations to the query locations.

The input encoder first uses a point-wise MLP that is shared across all locations to lift input features into high-dimensional encodings $\mathbf{f}^{(0)}$ and the encodings are then fed into stacked self-attention blocks. The update protocol inside each self-attention block is similar to the standard Transformer (Vaswani et al., 2017):

$$\mathbf{f}^{(l')} = \text{LayerNorm}\left(\mathbf{f}^{(l)} + \text{Attn}(\mathbf{f}^{(l)})\right), \quad \mathbf{f}^{(l+1)} = \text{LayerNorm}\left(\mathbf{f}^{(l')} + \text{FFN}(\mathbf{f}^{(l')})\right), \tag{6}$$

where $\text{Attn}(\cdot)$ denotes the linear attention mechanisms discussed in the previous sub-section, $\text{LayerNorm}(\cdot)$ denotes the layer normalization (Ba et al., 2016). When input and output data are not normalized, we drop the LayerNorm to allow the scaling to propagate through layers.

The query encoder consists of a shared point-wise MLP, whose first layer is a random Fourier projection layer (Tancik et al., 2020; Rahimi & Recht, 2007). The random Fourier projection $\gamma(\cdot)$ using Gaussian mapping is defined as:

$$\gamma(\mathbf{Y}) = [\cos{(2\pi\mathbf{YB})}, \sin{(2\pi\mathbf{YB})}], \tag{7}$$

where $\mathbf{Y} = [\mathbf{y}_1, \mathbf{y}_2, \dots, \mathbf{y}_n]^T$, $\mathbf{y}_i$ is the Cartesian coordinates of the $i$-th query point, $\mathbf{B} \in \mathbb{R}^{d_1 \times d_2}$ (with $d_1$ the dimension of input coordinates and $d_2$ the output dimension) is a matrix with elements sampled from Gaussian distribution $\mathcal{N}(0, \sigma^2)$ with predefined $\sigma$. The random Fourier projection $\gamma(\cdot)$ alleviate the spectral bias of coordinate-based neural networks (Tancik et al., 2020; Mildenhall et al., 2020), which has also been introduced into physics-informed machine learning in Wang et al. (2021c).

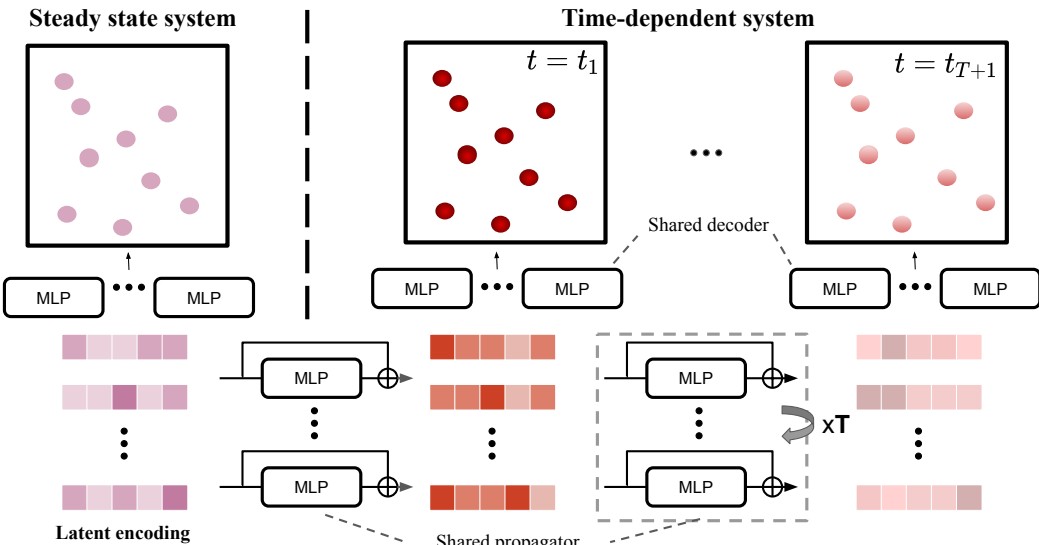

Figure 2: Decoder and propagator architecture. Given the latent encoding, for steady state system, a point-wise MLP is used to map the latent encoding $\mathbf{z}$ to the output function value. For time-dependent system, a point-wise MLP with skip connection is used to march states temporally in the latent space.

At the cross-attention module, by attending the latent encoding of the last ($L$-th) attention layer's output $\mathbf{f}^{(L)}$ of input function with the learned encoding $\mathbf{z}^{(0)}$ of query locations, we pass the information of input function to the query points. The cross-attention process is defined as:

$$\mathbf{z}' = \mathbf{z}^{(0)} + \text{Cross-Attn}(\mathbf{z}^{(0)}, \mathbf{f}^{(L)}), \quad \mathbf{z} = \mathbf{z}' + \text{FFN}(\mathbf{z}'). \tag{8}$$

Compared to equation 6, we drop the LayerNorm at each sub-layer to preserve the scale (variance) of $\mathbf{z}^{(0)}$.

**Latent propagator for time-dependent system**   A direct way to approach the time-dependent problem would be by extending the degrees of freedom and introducing temporal coordinates for each grid point as Raissi et al. (2019), which requires the model to learn the entire space-time solution at once. As analyzed in Krishnapriyan et al. (2021), for certain problems such as reaction-diffusion equation, this could lead to poor performance. In addition, this usually requires the model to have larger number of parameters in order to model the extra degree of freedom (Li et al., 2021a; Gupta et al., 2021).

Another direction would be using a time-marching scheme to autoregressively solve the entire state space with a fixed time discretization (resembles the *Method of lines* (Schiesser, 1991)), which is adopted by many neural-network based solvers as it is easier to train. A common architecture of learned time-marching solvers is the *Encoder-Processor-Decoder* (EPD) architecture (Sanchez-Gonzalez et al., 2020; Pfaff et al., 2020; Brandstetter et al., 2022; Stachenfeld et al., 2022), which takes state variable $u_t$ as input, project it into latent space, then processes the encoding with neural networks and finally decodes it back to the observable space: $\hat{u}^{t+\Delta t}$. Such model is trained by minimizing the per-step prediction error, with input $u^t$ and target $u^{t+\Delta t}$ sampled from ground truth data. In fact, if we unroll the learned per-step solver $f_{\text{NN}}(\cdot)$ to predict the full trajectory ($\hat{u}^1 = f_{\text{NN}}(u^0), \hat{u}^2 = f_{\text{NN}}(\hat{u}^1), \ldots$), the unrolled solver can be viewed as a recurrent neural network (RNN) with no hidden states, and thus the standard per-step training pipeline is essentially a recast of the *Teacher Forcing* (Williams & Zipser, 1989). Therefore, the learned per-step solvers face similar problems as RNNs trained with Teacher Forcing, the accumulated prediction error leads to the shift of the input data distribution and thus makes model perform poorly. Such data distribution shift can be alleviated by injecting noise during training (Sanchez-Gonzalez et al., 2020; Pfaff et al., 2020; Stachenfeld et al., 2022). Alternatively, Brandstetter et al. (2022) propose to unroll the solver for several steps and let the gradient flow through the last step to train the model on the shifted data. For operator learning problem where time horizon is fixed, Li et al. (2021a) push this further by fully unrolling the trajectory during training. However, fully unrolled training has a memory complexity of $O(tn)$ (with $t$ the length of time horizon and $n$ the number of model parameters), which can be prohibitively expensive if the model is large.

Here we approach the space-time modeling as a sequence-to-sequence (Sutskever et al., 2014) learning task and propose a recurrent architecture to propagate the state in the time dimension. Different from the standard EPD model, we forward the dynamics in the latent space instead of the observable space, and thus encoder only needs to be called once throughout the whole process, which reduces the memory usage and allows scalable fully unrolled training. Given the latent encoding $\mathbf{z}$ from equation 8, we set it as the initial condition $\mathbf{z}^0$ and recurrently forward the latent state with a propagator $N(\cdot)$ that predicts the residuals (He et al., 2015) between each step: $\mathbf{z}^{t+1} = N(\mathbf{z}^t) + \mathbf{z}^t$. There are many suitable choices for parametrizing the propagator (e.g. self-attention, message-passing layer), but in practice, we found a point-wise MLP shared across all the query locations and all the time steps suffice, which indicates that the our model can effectively approximate the original time-dependent PDE with a fixed-interval ODE in the latent space.

After reaching target time step in the latent space, we use another point-wise MLP to decode the latent encodings $z^t(x)$ back to the output function values $u(x,t)$.

**Relative positional encoding**  The attention mechanism is generally position-agnostic if no positional information is present in the input features. Galerkin/Fourier Transformer (Cao, 2021) approaches this problem by concatenating the coordinates into input features and latent embedding inside every attention head, and adds spectral convolutional decoder (Li et al., 2021a) on top of the attention layers. Instead of using convolutional architecture which is usually tied to the structured grid, we use Rotary Position Embedding (RoPE) (Su et al., 2021) to encode the positional information, which is originally proposed for encoding word's relative position.

Given the 1D Cartesian coordinate $x_i$ of a point and its $d$-dimensional embedding vector $\mathbf{q}_i \in \mathbb{R}^{d\times 1}$, the RoPE $\psi(\cdot)$ is defined as:

$$\psi\left(\mathbf{q}_i, x_i\right) = \mathbf{\Theta}\left(x_i\right)\mathbf{q}_i, \quad \mathbf{\Theta}\left(x_i\right) = \text{Diag}(\mathbf{R}_1, \mathbf{R}_2, \ldots, \mathbf{R}_{d/2}), \tag{9}$$

$$\text{where:} \quad \mathbf{R}_l = \begin{bmatrix} \cos\left(\lambda x_i \theta_l\right) & -\sin\left(\lambda x_i \theta_l\right) \\ \sin\left(\lambda x_i \theta_l\right) & \cos\left(\lambda x_i \theta_l\right) \end{bmatrix}, \tag{10}$$

where $\text{Diag}(\cdot)$ denotes stacking sub-matrices along the diagonal, $\lambda$ is wavelength of the spatial domain (e.g. $\lambda = 2048$ for a spatial domain discretized by 2048 equi-distant points), $\theta_l$ is set to $10000^{-2(l-1)/d}, l \in \{1, 2, \ldots, d/2\}$ following Vaswani et al. (2017); Su et al. (2021). Leveraging the property of the rotation matrix, the above positional encoding injects the relative position information when attending a query vector $\mathbf{q}_i$ with a key vector $\mathbf{k}_j$: $\psi\left(\mathbf{q}_i, x_i\right) \cdot \psi\left(\mathbf{k}_j, x_j\right) = (\mathbf{\Theta}(x_i)\mathbf{q}_i) \cdot (\mathbf{\Theta}(x_j)\mathbf{k}_j) = \mathbf{q}_i^T \mathbf{\Theta}(x_i - x_j)\mathbf{k}_j$. RoPE can be extended to arbitrary degrees of freedom as proposed in (Biderman et al., 2021) by splitting the query and key vectors. Suppose the spatial domain is 2D, the attention of query $\mathbf{q}_i$ with coordinate $\mathbf{x}_i = (\alpha_i, \beta_i)$ and key $\mathbf{k}_j$ with $\mathbf{x}_j = (\alpha_j, \beta_j)$ write as:

$$\begin{aligned} \psi\left(\mathbf{q}_i, \mathbf{x}_i\right) \cdot \psi\left(\mathbf{k}_j, \mathbf{x}_j\right) = \\ \psi_1\left((\mathbf{q}_i)_{:d/2}, \alpha_i\right) \cdot \psi_1\left((\mathbf{k}_j)_{:d/2}, \alpha_j\right) + \psi_2\left((\mathbf{q}_i)_{d/2:}, \beta_i\right) \cdot \psi_2\left((\mathbf{k}_j)_{d/2:}, \beta_j\right), \end{aligned} \tag{11}$$

where $(\mathbf{q}_i)_{:d/2}$ denotes the first $d/2$ dimensions of vector $\mathbf{q}_i$ and $(\mathbf{q}_i)_{d/2:}$ denotes the rest $d/2$ dimensions, similarly for $\mathbf{k}_j$, and $\psi_1(\cdot), \psi_2(\cdot)$ is the 1D RoPE as defined in equation 9.

**Training setting**  The general training framework of this work is similar to the previous data-driven operator learning models (Gupta et al., 2021; Li et al., 2021a; Cao, 2021; Anandkumar et al., 2020; Lu et al., 2021a). Given the input function $a \in \mathcal{A}$, and target function $u \in \mathcal{U}$, our goal is to learn an operator $\mathcal{G}_\theta : \mathcal{A} \mapsto \mathcal{U}$. The model parameter $\theta$ is optimized by minimizing the empirical loss $\mathbb{E}_{a \sim \mu}\left[\mathcal{L}(\mathcal{G}_\theta(a), u)\right]$, where $\mu$ is a measure supported on $\mathcal{A}$, and $\mathcal{L}(\cdot)$ is the appropriate loss function. In practice, we evaluate this loss numerically based on the sampling of $\{a_{D(x)}^{(j)}, u_{D(y)}^{(j)}\}_{j=1}^N$ on discrete grid points $D(x) : \{x_i\}_{i=1}^n, D(y) : \{y_i\}_{i=1}^m$, and choose $\mathcal{L}(\cdot)$ as relative $\mathcal{L}_2$ norm: $\frac{1}{B}\sum_{j=1}^B \frac{\|\hat{u}^{(j)} - u^{(j)}\|_2}{\|u^{(j)}\|_2}$, given $B$ samples with $\hat{u}^{(j)}$ being the model prediction. Different from several previous works (Li et al., 2021a; 2020b; Cao, 2021; Gupta et al., 2021), $D(x), D(y)$ does not need to be the same discretization of the PDE spatial domain $\Omega$ and can vary across different training samples $\{a_{D(x)}^{(j)}, u_{D(y)}^{(j)}\}$.

## 4 Experiment

In this section, we first present the benchmark results on the standard neural operator benchmark problems from Li et al. (2021a) and compare our model with several other state-of-the-art operator learning frameworks, including Galerkin/Fourier Transformer (G.T. / F.T.) from Cao (2021) and Multiwavelet-based Operator (MWT) (Gupta et al., 2021). Then we showcase the model's application to irregular grids where above frameworks cannot be directly applied to, and compare model to a graph neural network baseline (Lötzsch et al., 2022). We also perform an analysis on the model's latent encoding. Finally, we present an ablation study of key architectural choices. (The full details of model architecture on different problems and training procedure are provided in the Appendix A. More ablation studies of hyperparameter choices are provided in Appendix C.)

### 4.1 Benchmark problems

Below we provide a brief description of investigated PDEs. Full details of these equations' definition and dataset generation are provided in the Appendix D.

**Problems on uniform grid**   The first kind of problems considered include 1D viscous Burgers' equation, 2D Darcy flow, and 2D imcompressible Navier-Stokes equation. In these problems, the data are represented on a uniform equi-spaced grid, where discrete Fourier transformation (FNO, G.T./F.T.) and wavelet transformation (MWT) can be efficiently applied to.

The objective solution operators $\mathcal{G}$ aimed to learn are defined as ($T$ is the time horizon, $u$ denotes solution function of interest, $a$ denotes the coefficient function):

$$
\begin{align}
\textit{Burgers':}\quad & \mathcal{G}: u(\cdot,t)|_{t=0} \mapsto u(\cdot,t)|_{t=1}, \tag{12}\\
\textit{Darcy flow:}\quad & \mathcal{G}: a \mapsto u, \tag{13}\\
\textit{Navier-Stokes:}\quad & \mathcal{G}: u(\cdot,t)|_{t\in[0,10]} \mapsto u(\cdot,t)|_{t\in(10,T]}. \tag{14}
\end{align}
$$

On Navier-Stokes equation, following prior work (Li et al., 2021a), we adopt 3 datasets NS1, NS2, NS3 with different viscosities, and we generate a more challenging and diverse dataset (denoted as NS-mix) where $\nu$ is not constant, and it is uniformly sampled: $\nu \sim \mathcal{U}([0.4,1]) \times 10^{-4}$, corresponding to Reynolds numbers roughly ranging from 200 to 500. For the size of each Navier-Stokes dataset, NS2-full contains 9800/200 (train/test) samples; NS-mix is a larger dataset consisting of 10000/1000 samples; each of the rest datasets contains 1000/200 samples. For Burgers' equation and Darcy flow, the dataset splitting is same as in Cao (2021).

**Problems on non-uniform grid**   Other than uniform grid, the second kind of problems considered are based on unstructured non-uniform grid. We first investigate the 2D Poisson equation of magnetic/electric field on different geometries, the objective is to find the solution of field given boundary conditions and source-terms. We use the shape extrapolation dataset from Lötzsch et al. (2022). The training set contains grids of four shapes (square, circle with or without hole, L-shape) with 8000 samples, while testing set has 2000 samples consisting of U-shape grids that model has never seen during training. This experiment tests model's generalization capability to different solution domain. In addition, we study the 2D time-dependent compressible flow around the cross-section of airfoils using the dataset from Pfaff et al. (2020). In this problem, the underlying grid is highly irregular, as the distance between two closest points ranges from $2e-4$ to 3.5. The dataset contains 1000/100 sequences of different inflow speed (Mach number) and angles of attack.

The objective solution operators $\mathcal{G}$ with respect to these problems are defined as:

$$
\begin{align}
\textit{Poisson:}\quad & \mathcal{G}: f \mapsto u, \quad u=0 \text{ on } \partial\Omega, \tag{15}\\
\textit{Airfoil:}\quad & \mathcal{G}: \mathbf{u}(\cdot,t)|_{t\in[0,0.576]} \mapsto \mathbf{u}(\cdot,t)|_{t\in(0.576,4.800]}, \tag{16}
\end{align}
$$

where $\Omega$ is the solution spatial domain and $f$ is the source term of the Poisson equation. For Poisson equation we predict the field vector in addition to the scalar potential [2]. For compressible flow we predict velocity, density and pressure jointly.

## 4.2 Results and discussion

For benchmark results of all baselines, we report the evaluation results from original paper when applicable. When not applicable, we train and evaluate the model following the suggested setting in the original paper, and mark the results with "*". The best version of Galerkin/Fourier Transformer is reported for each problem. Details of baseline models' implementation and comparison of calculation time can be found in the Appendix B.

| Data settings | | Relative $L_2$ norm | | | | |
|---|---|---|---|---|---|---|
| Case | $\nu, T$ | FNO-2D | FNO-3D | MWT | G.T.* | OFormer |
| NS1 | $1 \times 10^{-3}, 50$ | 0.0128 | 0.0086 | **0.0062** | 0.0098 | $0.0104 \pm {\scriptstyle 0.0005}$ |
| NS2-part | $1 \times 10^{-4}, 30$ | 0.1559 | 0.1918 | 0.1518 | **0.1399** | $0.1755 \pm {\scriptstyle 0.0059}$ |
| NS2-full | $1 \times 10^{-4}, 30$ | 0.0834 | 0.0820 | 0.0667 | 0.0792 | $\mathbf{0.0645} \pm {\scriptstyle 0.0011}$ |
| NS3 | $1 \times 10^{-5}, 20$ | 0.1556 | 0.1893 | 0.1541 | **0.1340** | $0.1705 \pm {\scriptstyle 0.0007}$ |
| NS-mix* | $[0.4, 1] \times 10^{-4}, 30$ | 0.1650 | 0.1654 | **0.1267** | 0.1462 | $0.1402 \pm {\scriptstyle 0.0016}$ |
| # of parameters (M) | | 0.41/2.37 | 6.56 | 179.18 | 1.56 | 1.85 |

Table 1: Benchmark on 2D Navier-Stokes equation with fixed $64 \times 64$ grid. For large dataset (NS-mix) we use a larger version of FNO-2D by increasing modes and width. We adopt Galerkin Transformer (G.T.) with a larger hidden dimension ($64 \to 96$) to match the size of other models.

| Res. | Relative $L_2$ norm ($\times 10^{-3}$) | | | | |
|---|---|---|---|---|---|
| | FNO* | FNO+* | MWT | F.T. | OFormer |
| 512 | 3.15 | **0.72** | 1.85 | 1.14 | $1.42 \pm {\scriptstyle 0.03}$ |
| 2048 | 3.07 | **0.72** | 1.86 | 1.12 | $1.30 \pm {\scriptstyle 0.07}$ |
| 8192 | 2.76 | **0.70** | 1.78 | 1.07 | $1.54 \pm {\scriptstyle 0.10}$ |

Table 2: Benchmark on 1D Burgers' equation with different resolution. FNO uses ReLU (Nair & Hinton, 2010) whereas FNO+ uses GELU. Both FNO variants remove normalization compared to the original paper.

| Res. | Relative $L_2$ norm ($\times 10^{-2}$) | | |
|---|---|---|---|
| | FNO | G.T. | OFormer |
| 141 | 1.09 | **0.84** | $1.26 \pm {\scriptstyle 0.03}$ |
| 211 | 1.09 | **0.84** | $1.28 \pm {\scriptstyle 0.02}$ |

Table 3: Benchmark on 2D Darcy flow with different resolution. MWT's result is not reported since its implementation requires resolution to be power of 2 and thus it is trained on a different data.

**Performance on equi-spaced grid** The comparison results with several state-of-the-art baselines are shown in Table 1, 2, 3. We observe that for small data regime (NS2-part, NS3) and problems with relatively smooth target function (viscid Burgers', Darcy), OFormer generally does not show superiority over other operator learning-frameworks. Architectures like spectral convolution (in FNO, Galerkin/Fourier Transformer) are baked in helpful inductive bias when target is smooth and periodic. As shown in Table 2, equipped with a smooth activation function, FNO+ outperforms all other methods significantly, as the target function $u(\cdot, 1)$ is relatively smooth due to the presence of diffusion term, and has a periodic characteristic in the spatial domain. For more complex instances like Navier-Stokes equation, we find that OFormer has strong performance in data sufficient regime, where it is on par with MWT on NS2-full with substantially less parameters. This highlights the learning capacity of attention-based architecture, which leverages flexible basis learned from data. Moreover, our model performs similarly under different resolutions, indicating its robustness to different resolution. (The temporal error trend can be found in Appendix C. Contour plot of the prediction and corresponding error can be found in Appendix E.)

---

[2]The electric field $\mathbf{E}$ is defined as the negative gradient of electric potential ($\phi$): $\mathbf{E} = -\nabla\phi$. For magnetic field $\mathbf{B}$, it is defined as the curl of magnetic potential $\mathbf{A}$: $\mathbf{B} = \mathrm{rot}\mathbf{A}$. In this problem $A_x = A_y = 0$, and therefore magnetic potential reduces to: $B_x = \partial A_z/\partial y, B_y = -\partial A_z/\partial x$. In the model we use a separate head to predict these fields.

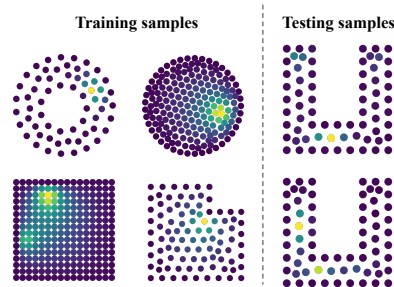

Figure 3: Visualization of training/testing samples in 2D Poisson problem. Testing set contains new geometries that never appeared in the training set.

| Mesh aug. | Model | Mean squared error ($\times 10^{-3}$) | | | |
|---|---|---|---|---|---|
| | | Electric | | Magnetic | |
| | | Potential | Field | Potential | Field |
| No | OFormer | $\mathbf{0.174} \pm {}_{0.083}$ | $\mathbf{1.690} \pm {}_{0.201}$ | $\mathbf{0.138} \pm {}_{0.021}$ | $\mathbf{1.470} \pm {}_{0.172}$ |
| | GNN | 0.327 | 2.723 | 0.161 | 2.640 |
| Yes | OFormer | $0.019 \pm {}_{0.004}$ | $\mathbf{0.328} \pm {}_{0.025}$ | $0.016 \pm {}_{0.002}$ | $\mathbf{0.195} \pm {}_{0.009}$ |
| | GNN | $\mathbf{0.006}$ | 1.821 | $\mathbf{0.006}$ | 1.338 |

Table 4: Benchmark on electrostatics and magnetostatics' Poisson equation. The train/evaluation protocol is adopted from Lötzsch et al. (2022). Mesh augmentation comprises a set of random transformation on the training meshes, including varying node density and size of hole/cutout.

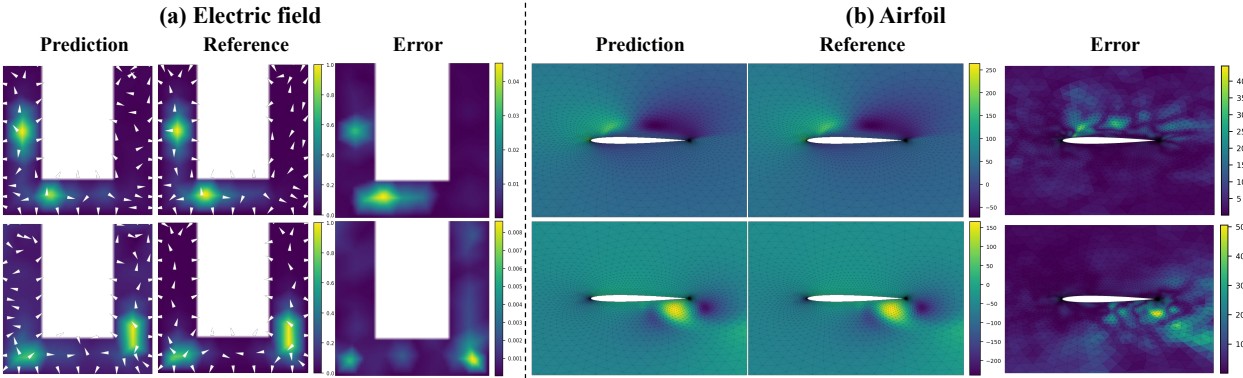

Figure 4: Samples of model's prediction, reference simulation result and absolute error on: **(a)** Electric potential and corresponding field; **(b)** Velocity field around airfoil, top/bottom: x/y component.

**Application to irregular grids** The comparison between OFormer and GNN baseline on the 2D Poisson problem is shown in Table 4. Both OFormer and GNN have the capability to learn and extrapolate to new geometries with reasonable accuracy under mesh augmentation. However, we can observe that OFormer outperforms GNN by a margin when no mesh augmentation is applied. This highlights an essential difference in the learning mechanism between OFormer and local graph convolution-based architecture. GNN that exploits local mesh information such as edge distance and node connection is more prone to overfit on the training geometries, while OFormer, leveraging global attention and not using any explicit graph information, is more robust under the change of local mesh topology. Other than grid with relatively uniform point distribution, OFormer is also applicable to highly irregular meshes. We showcase that OFormer is able to predict the temporal evolution of compressible flow around the airfoil (exemplar visualization is shown in Figure 4, more visualization results can be found in Appendix E). The test root mean squared error (at region closed to airfoil) of the predicted momentum is $15.469 \pm 0.074$ (relative error: $8.17 \pm 0.06\%$).

Another important property of OFormer is that it can handle variable number of input/query points with the same set of weights. Here we showcase an application of such property. As shown in Figure 5, the model can still predict reasonably accurate results on a full-resolution output grid based on sparse input. The input is generated by randomly sampling 25% of the grid points in the spatial domain, which is 150% of the maximum dropping ratio during training[3]. (More quantitative results on varying sampling grids can be found in Appendix C.)

---

[3]During training, we randomly drop some of the input points with a ratio of $r \sim \mathcal{U}([0, 0.5])$, but still query at full resolution and train on full resolution.

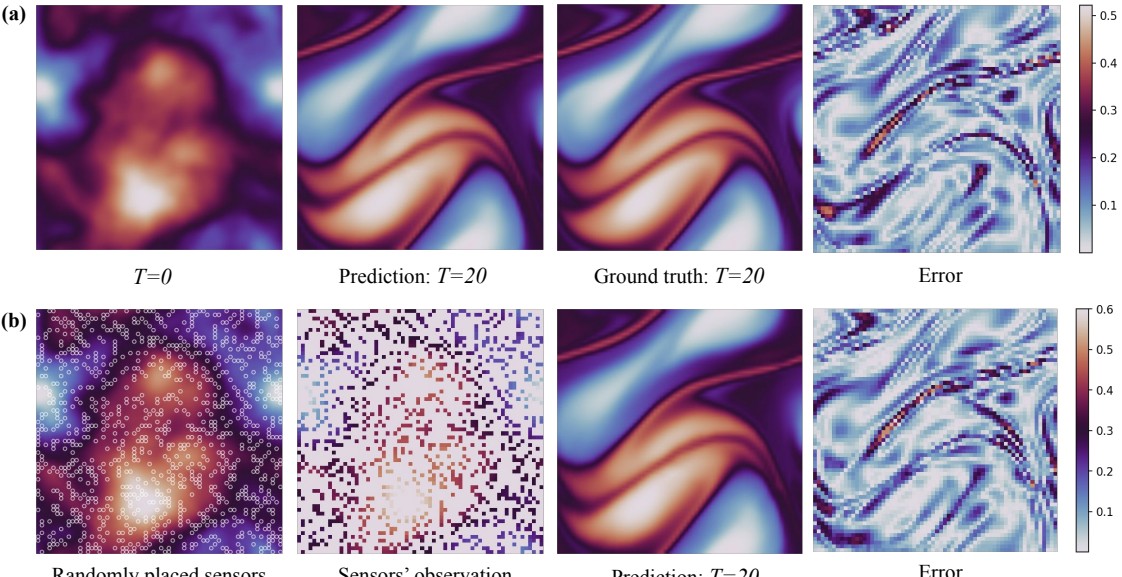

Figure 5: **(a)** Model's prediction on dense uniform input points; **(b)** Model's prediction on sparse randomly sampled input points (25% of all grid points).

**Learned spatio-temporal representation**   To inspect OFormer's ability to learn meaningful representations of the PDEs, we perform an analysis on our model trained on a dataset of Navier-Stokes equation where each sample has a random coefficient of $\log(\nu) \sim \mathcal{U}([-3, -5.3])$, which resulted in $\nu$ falls in the range of $(5e-6, 1e-3)$. The model is trained to predict the future time steps, where it has never been informed of the coefficient values. After training, we examine the latent space of the input encoder.

A single $d$-dimensional representation vector $\mathbf{f}_{\text{avg}}^{(k)}$ of $k$-th sample is obtained by applying average pooling on the encodings $\mathbf{f}_{\text{avg}}^{(k)} = \text{Avg}(\{\mathbf{f}_i^{(k)}\}_{i=1}^n)$ over the input points. First, we project the representation vectors of all training samples into a 2-dimensional space using Principal Component Analysis (PCA). We can observe the evident trend of colors w.r.t. $\nu$ in Figure 6(a). Furthermore, we train a linear regression model on the latent vectors from the training samples, and then we use it to predict the $\nu$ on testing samples. Figure 6(b) shows that a linear model can effectively map the latent representations to the viscosity coefficients which indicates the model's capability of finding compact representations such that a nonlinear mapping has become linear.

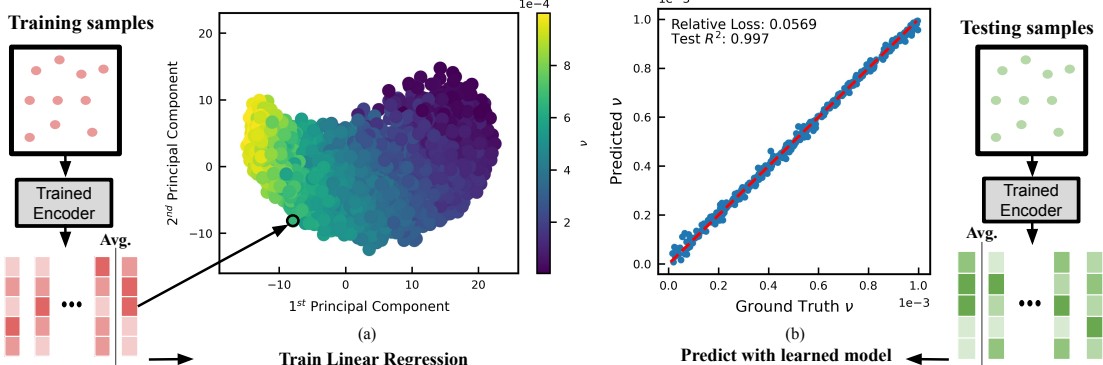

Figure 6: Learned representations of the model trained on Navier-Stokes equations with different viscosity coefficients. **(a)** The 2-dimensional PCA on the latent vectors. **(b)** A linear regression model trained on the latent vectors of training samples can accurately predict $\nu$ of testing samples.

## 4.3 Ablation study

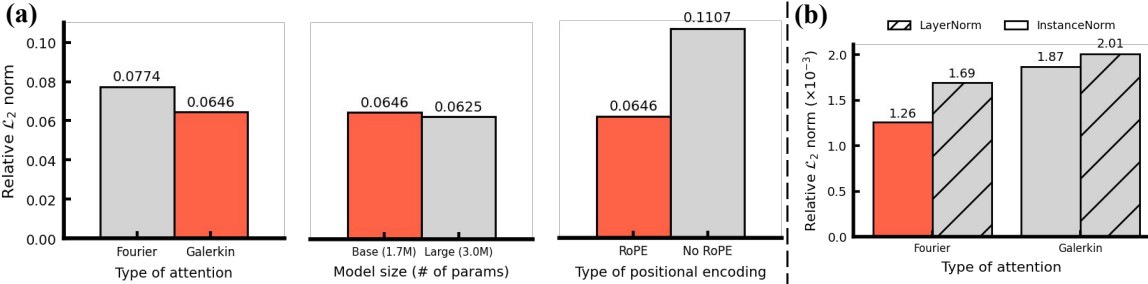

Figure 7: Ablation study on **(a)** NS2-full dataset; **(b)** 1D Burgers' equation with resolution 2048. Light color denotes option we adopted.

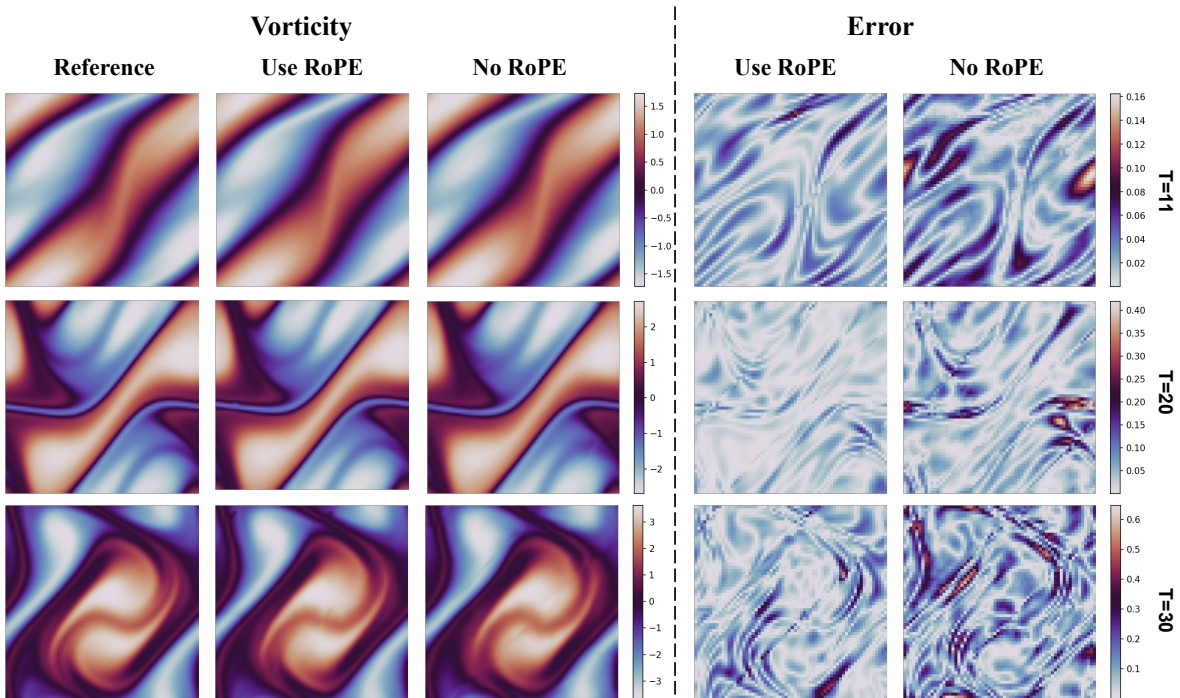

Figure 8: Influence of positional encoding, the visualized sequence is randomly chosen. Error is measured by absolute error.

We study the influence of the following factors on the model's performance: (i) The type of attention; (ii) Model's scaling performance; (iii) Positional encoding; (iv) Different normalization methods. The quantitative ablation results are shown in Figure 7.

First of all, we find that the performance of a specific type of attention is problem-specific, which signifies the importance of choosing normalization targets (query/key or key/value). In addition, we observe that on a scale-sensitive problem (data are not normalized for Burgers' equation to preserve the energy decaying property), InstanceNorm which normalizes each column vector $||\mathbf{v}^j||_2 = 1$, performs better than LayerNorm, further facilitating the bases interpretation of two linear attention mechanisms. Scaling up the model's size (increasing the hidden dimension) can benefit the model's performance in a data sufficient regime. On NS-full dataset, the larger version of OFormer pushes the previous best result from 0.0667 (Gupta et al., 2021) to 0.0625. Lastly, we find that if RoPE is removed and replaced by concatenating coordinates into query and key vectors (after projection), the model's performance deteriorates significantly. As shown in Figure 8, without

Rotary Position Embedding (RoPE), the model's prediction becomes relatively blurrier and generally has larger error. This signifies the importance of relative position information in PDE operator learning.

## 5 Conclusion

In this work, we introduce OFormer, a fully point-based attention architecture for PDEs' solution operator learning. The proposed model shows competitive performance on PDE operator learning and offers flexibility with input/output discretizations. We also showcase that a trained data-driven model can learn to generalize to diverse instances of PDEs without inputting system parameters (NS-mix) and learn meaningful representation for system identification. The major limitation is that given few assumptions are made on the grid structure, the model needs sufficient data to reach its optimal performance, and linear attention mechanism is still relatively computationally expensive for higher resolution grids. In addition, just like most of the other data-driven models, the accumulated error of the model's prediction on long-term transient phenomena such as turbulent fluids is unneglectable, unlike numerical solvers which are usually guaranteed to be stable.

## 6 Acknowledgments

This work is supported by the start-up fund from the Department of Mechanical Engineering, Carnegie Mellon University, United States.

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

## Appendix Summary

- Section A: Model implementation details;
- Section B: Baseline details;
- Section C: Further ablation study;
- Section D: Dataset details;
- Section E: Further results and visualization;

## A Model implementation details

Below we provide the full implementation details of models used in different problems. All the models are implemented in PyTorch (Paszke et al., 2019).

For notation, we use $[d_1, d_2]$ to denote a linear/hidden layer with input dimension $d_1$ and output dimension $d_2$, which corresponds to nn.Linear(d1, d2) in PyTorch. For attention layer, $d \times h$ indicates that there are $h$ attention heads each with a dimension of $d$.

### A.1 Hyperparameter for equi-spaced grid problem

| Problem | Input encoder top | SA hidden dim | SA FFN | # SA Block | Input encoder bottom |
|---|---|---|---|---|---|
| 1D Burgers | [2, 96] | 96×1 | [96, 192, 96] | 4 | [96, 96] |
| 2D Darcy flow | [3, 96] | 96×4 | [96, 192, 96] | 4 | [96, 256] |
| 2D Navier-Stokes | [12, 128] | 128×1 | [128, 256, 128] | 5 | [128, 192] |

Table 5: Architecture of input encoder. SA denotes self-attention, FFN denotes feed forward network.

| Problem | Query encoder top | CA hidden dim | CA FFN | Query encoder bottom |
|---|---|---|---|---|
| 1D Burgers | [1, 96, 96] | 96×8 | [96, 192, 96] | - |
| 2D Darcy flow | [2, 256, 256] | 256×4 | [256, 512, 256] | - |
| 2D Navier-Stokes | [2, 192, 192] | 192×4 | [192, 384, 196] | [192, 384] |

Table 6: Architecture of query encoder. CA denotes cross-attention.

| Problem | Propagator | Decoder | Total # of params (M) |
|---|---|---|---|
| 1D Burgers | [96, 96, 96, 96]×3 | [96, 48, 1] | 0.67 |
| 2D Darcy flow | - | [256, 128, 64, 1] | 2.51 |
| 2D Navier-Stokes | [384, 384, 384, 384] | [384, 192, 96, 1] | 1.85 |

Table 7: Architecture of propagator and decoder. The number of parameters is the total number including input/query encoders. For 1D Burgers' equation we use 3 unshared MLPs for propagating the dynamics.

### A.2 Hyperparameter for irregular grid problem

We use GELU (Hendrycks & Gimpel, 2016) as activation function for propagator and decoder, and opt for Gated-GELU (Dauphin et al., 2016) for FFN. On 1D Burgers, we train the model for 20k iterations using a batch size of 16. On Darcy flow, we train the model for 32K iterations using a batch size of 8. For small dataset on Navier-Stokes (consisting of 1000 training samples), we train the model for 32k iterations using a batch size of 16, and 128k iterations for large dataset (consisting of roughly 10k training samples). On Electrostatics/Magnetostatics we train the model for 32k iterations using a batch size of 16. On Airfoil we train model for 50k iterations using a batch size of 10.

| Problem | Input encoder top | SA hidden dim | SA FFN | # SA Block | Input encoder bottom |
|---|---|---|---|---|---|
| 2D Electrostatics | [11, 64, 64] | 64×1 | [64, 128, 64] | 2 | [64, 64] |
| 2D Magnetostatics | [11, 96, 96] | 96×1 | [96, 128, 96] | 2 | [96, 96] |
| 2D Airfoil | [6, 128, 128]* | 128×1 | [128, 256, 128] | 5 | [128, 128] |

Table 8: Architecture of input encoder. The input of 2D airfoil contains an additional time dimension of length $t_{in}$, so the very top layer is a Conv2D layer with kernel size $(t_{in}, 1)$ stride $(t_{in}, 1)$ and no padding.

| Problem | Query encoder top | CA hidden dim | CA FFN | SA hidden dim | Query encoder bottom |
|---|---|---|---|---|---|
| 2D Electrostatics | [3, 64, 64] | 64×4 | [64, 128, 64] | 64×1 | - |
| 2D Magnetostatics | [3, 96, 96] | 96×4 | [96, 192, 96] | 96×1 | - |
| 2D Airfoil | [2, 128, 128]* | 128×4 | [128, 256, 128] | 128×1 | [128, 256] |

Table 9: Architecture of query encoder. On non-uniform grid, we add a self-attention layer after cross-attention layer. For 2D airfoil, we introduce a learnable embedding for each node type (boundary, open area, airfoil)

| Problem | Propagator | Decoder | Total # of params (M) |
|---|---|---|---|
| 2D Electrostatics | - | [64, 32, 32, 3] | 0.17 |
| 2D Magnetostatics | - | [96, 48, 48, 3] | 0.31 |
| 2D Airfoil | [384, 256, 256, 256] | [256, 128, 128, 4] | 1.35 |

Table 10: Architecture of propagator and decoder. The number of parameters is the total number including input/query encoders. For airfoil, before inputting into propagator block, node type embedding are concatenated together with other features.

For training on the 2D Navier-Stokes and Airfoil problem, we adopt a curriculum strategy to grow the prediction time steps. During the initial stage, instead of predicting all the future states towards the end of time horizon, e.g. $u_{t_0}, u_{t_1}, \ldots, u_T$, we truncate the time horizon with a ratio $\gamma < 1$ (heuristically we choose $\gamma \approx 0.5$), train the network to predict $u_{t_0}, u_{t_1}, \ldots, u_{\gamma T}$. The motivation is that at the initial stage the training dynamics is less stable and gradient is changing violently, we truncate the time horizon to avoid stacking very deep recurrent neural network and thus alleviate the unstable gradient propagation. In practice, we found this makes the training more stable and converge slightly faster.

As suggested in (Cao, 2021), for scaling-sensitive problem - the Burgers' equation, the initialization of the query/value projection layer has a great influence on the model's performance. We use orthogonal initialization (Saxe et al., 2013) with gain $1/d$ and add an additional constant $1/d$ to the diagonal elements (with $d$ being the dimension of latent dimension at each head). The orthogonal weights have the benefit of preserving the norm of input and we found that with proper scaling it provides better performance than standard Xavier uniform initialization. Please refer to the Appendix C for the study on the influence of initialization.

For most of the problems we studied, we use instance normalization in the attention and observe slight improvement compared to layer normalization. However, for 2D Electrostatics/Magnetostatics' Poisson equation, we find that instance normalization is unstable as the grid points in this problem are much fewer (60 - 250) than other problems, so we adopt layer normalization for this problem.

## B    Baseline details

We use the open-sourced official implementation of Fourier Neural Operator (FNO)[4] (Li et al., 2021a), Multiwavelet Operator (MWT)[5] (Gupta et al., 2021) and Galerkin/Fourier Transformer (G.T./F.T.)[6] (Cao, 2021) to carry out the benchmark. The experiments on these baselines are carried out following the settings in the official implementation with minimal changes.

For FNO, when applying to 2D Navier-Stokes problem with relatively large number of data samples, we increase the number of modes in Fourier Transformation to 12 and increase the width (hidden dimension of network) to 32. This notably improves FNO-2D's performance. For G.T./F.T., on 2D Navier-Stokes problem we increase the hidden dimension size from 64 to 96, and extend the training epochs from 100 to 200 (150 on large dataset with 10000 training samples).

Below we provide a computational benchmark of different models' training and inference efficiency on 2D Navier-Stokes equation. The benchmark was conducted on a RTX-3090 GPU using PyTorch 1.8.1 (1.7 for MWT) and CUDA 11.0. The number of training samples is 10000, with time horizon $T = 30$, and the batch size is set to 10.

| Model | Iters / sec | Memory (GB) | # of params (M) |
|---|---|---|---|
| FNO-2D | 9.71 | 0.33 | 2.37 |
| FNO-3D | 10.74 | 0.42 | 6.56 |
| MWT | 1.67 | 10.95 | 179.18 |
| G.T. | 1.79 | 16.65 | 1.56 |
| OFormer | 1.89 | 15.93 | 1.85 |

Table 11: Training computational cost of different models.

| Model | OFormer | G.T. | FNO-2D | FNO-3D | MWT | Pseudo-spectral | |
|---|---|---|---|---|---|---|---|
| | | | | | | Fine | Coarse |
| Time (sec) | 2.32 | 3.22 | 0.58 | 0.50 | 2.21 | 5735.74 | 3510.97 |

Table 12: Inference computational cost of different models and ground truth numerical solver. The time indicates how long does it take for each model to finish simulating 200 sequences (data processing time is excluded). The pseudo-spectral method is taken from (Li et al., 2021a), which is implemented in PyTorch. Fine grid has a resolution of $256 \times 256$ while coarse grid has a resolution of $64 \times 64$.

Note that as many traditional numerical solvers do not support GPU or not optimized for GPU computation, and they can trade-off accuracy for faster runtime (e.g. adopting a coarse grid and time step), so it is hard to conclude the exact acceleration of learned solvers on different problems, but in general we can observe that for time-dependent system learned solvers are highly efficient as they can tolerate a very large time step size (usually 3-4 orders of magnitudes compared to solver).

## C    Further ablation study

In this section we present further ablation study for the model.

**Sparse reconstruction and prediction**    We study the model's performance under different discretization settings. At the training stage, if some of the input points (with probability $p = 0.2$ and dropping ratio $r \sim \mathcal{U}([0, 0.5])$) are randomly dropped, the model's performance are observed to deteriorate on most of the datasets. This indicates that while attention is flexible with respect to the input discretization, its best performances are usually obtained by exploiting the very specific grid structures. But for dataset that contains more diverse system parameters, like NS-mix and NS-mix2, the random dropping has little influence

---

[4]https://github.com/zongyi-li/fourier_neural_operator
[5]https://github.com/gaurav71531/mwt-operator
[6]https://github.com/scaomath/galerkin-transformer

on model's performance. While models trained with random dropping usually tend to perform worse than models trained with fixed grid points, they are much more robust on irregular discretization as shown in the Table 14 and can still generate reasonable prediction based on 25% of the input points.

| Dataset | No random drop | Random drop | Performance diff. |
|---------|----------------|-------------|-------------------|
| NS1 | 0.0104 | 0.0133 | 0.0029 (27.9%) |
| NS2-part | 0.1702 | 0.1773 | 0.0071 (4.2%) |
| NS2-full | 0.0646 | 0.0672 | 0.0026 (4.0%) |
| NS3 | 0.1697 | 0.1747 | 0.0050 (2.9%) |
| NS-mix | 0.1400 | 0.1413 | 0.0013 (0.9%) |
| 1D Burgers | 0.00126 | 0.00142 | 0.00016 (12.7%) |

Table 13: Influence of randomly dropping some input grid points during the training process. No random drop indicates during training input and output grids are fixed at $64 \times 64$; random drop indicates that some are the input grid points are randomly dropped during training, while output grid is still fixed at $64 \times 64$. Error is measured by relative $\mathcal{L}_2$ norm. The resolution of Burgers' equation is 2048.

| Training strategy | All | 95% | 75% | 50% | 25% |
|-------------------|-----|-----|-----|-----|-----|
| Random drop | 0.0672 | 0.0687 | 0.0772 | 0.0939 | 0.1294 |
| No random drop | 0.0646 | 0.0795 | 0.1293 | 0.2065 | 0.3774 |

Table 14: Performance of models on NS2-full's test set with randomly sampled input points. Percentage on the top row indicates the ratio of sampled points' amount with respect to total number of all points. Error is measured by relative $\mathcal{L}_2$ norm on comparing full-resolution ($64 \times 64$) prediction with ground truth.

**Influence of initialization**   On Burgers' equation, to emulate the energy decaying scheme and preserve the magnitude of how much the energy has decayed, no data normalization or normalization layer (except for the instance normalization inside every attention layer) is used. While this improves model's performance, it also makes the model very sensitive to the initialization of projection matrix inside each attention layer, especially for the matrix where no instance normalization is applied (for Fourier type attention, it's the value matrix; for Galerkin type attention, it's the query matrix).

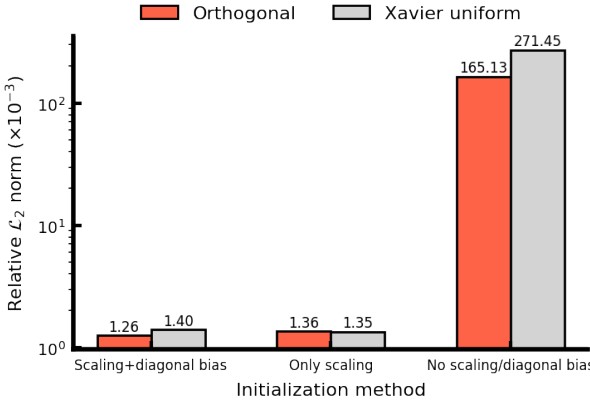

Figure 9: Influence of different initialization on scaling sensitive problem (1D Burgers' equation), using Fourier-type attention.

We modulate the initialization process of the weight of query (or value if using Fourier type attention) projection matrices ($\mathbf{W}_{(q)} \in \mathbb{R}^{d \times d}$) in Galerkin type attention as:

$$\mathbf{W}_{(q)} = \sigma \mathbf{B} + \delta \mathbf{I}, \tag{17}$$

where $\sigma$ is a scaling factor which we set to $1/d$, $\mathbf{B}$ is the original weight matrix (initialized either via orthogonal initialization (Saxe et al., 2013) or Xavier uniform (Glorot & Bengio, 2010)), and similar to Cao (2021) we add diagonal bias $\delta\mathbf{I}$ to the matrix. The orthogonal initialization initializes a weight matrix that is norm preserving and projects the input feature vectors onto orthogonal embedding vectors. As shown in Figure 9, due to presence of skip connection: $\mathbf{z}' = \mathbf{z} + \text{Attn}(\mathbf{z})$ and there is no normalization layer, both initialization methods blow up and have much worse performance when no scaling is applied. Orthogonal initialization tend to perform the best when both scaling (set gain to $1/d$) and diagonal bias are applied.

**Influence of data** OFormer's performance is highly correlated with the amount of data available, especially for relatively complicated problem like Navier-Stokes equation. As shown in Figure 10, the performance of the trained OFormer improves almost linearly as data size grow exponentially. At low data regime (500-5000), OFormer benefits significantly from increasing the data amount. This indicates that data is crucial for OFormer to unlock its optimal approximation power and also demonstrates OFormer's strong capacity to assimilate large data. However, this also hints a potential bottleneck of OFormer that it might not have satisfactory performance when there is limited data.

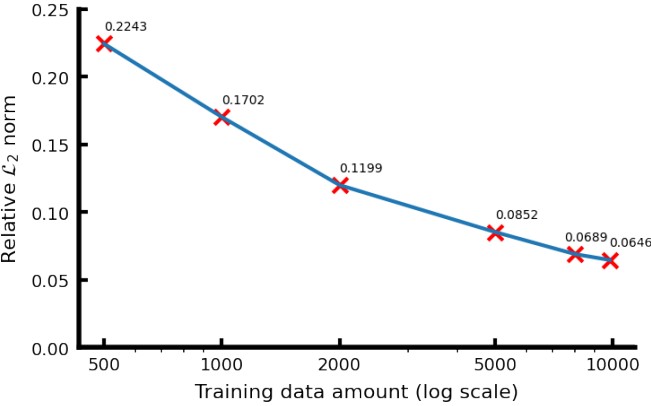

Figure 10: Performance of OFormer on NS2 when trained with different amount of data.

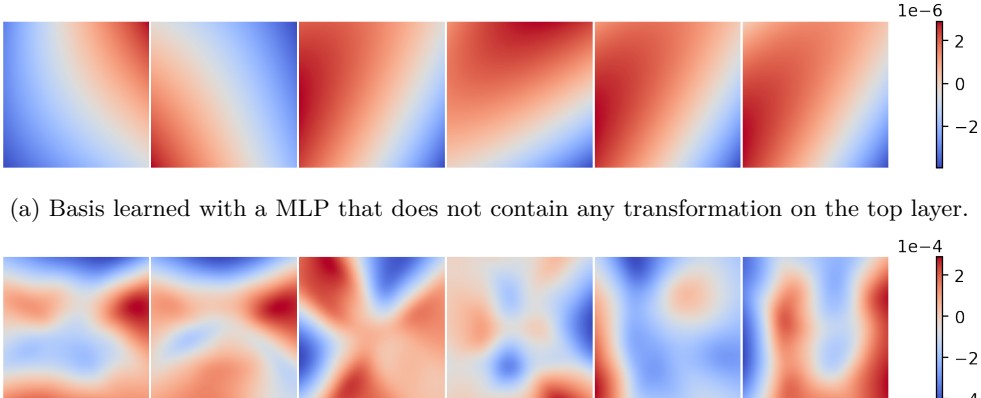

(a) Basis learned with a MLP that does not contain any transformation on the top layer.

(b) Basis learned with a MLP that has RFF on the top layer.

Figure 11: Visualization of bases learned (the output of coordinate-based network in the decoder part) on 2D Darcy flow. The visualized bases are randomly chosen.

**Influence of Random Fourier Feature** We study the influence of Random Fourier Features (RFF) (Tancik et al., 2020; Rahimi & Recht, 2007) on 3 different problems - 1D Burgers, 2D Darcy flow and Navier-Stokes. As shown in Figure 12, applying RFF to the input coordinates boosts the performance on all

three problems. We hypothesize the main reason is the original MLP tends to learn a set of bases with lower frequency while RFF broadens the spectrum of these earned bases (Tancik et al., 2020). As an example, Figure 11 reveals that without RFF, learned bases have a relatively smoother pattern than those with RFF.

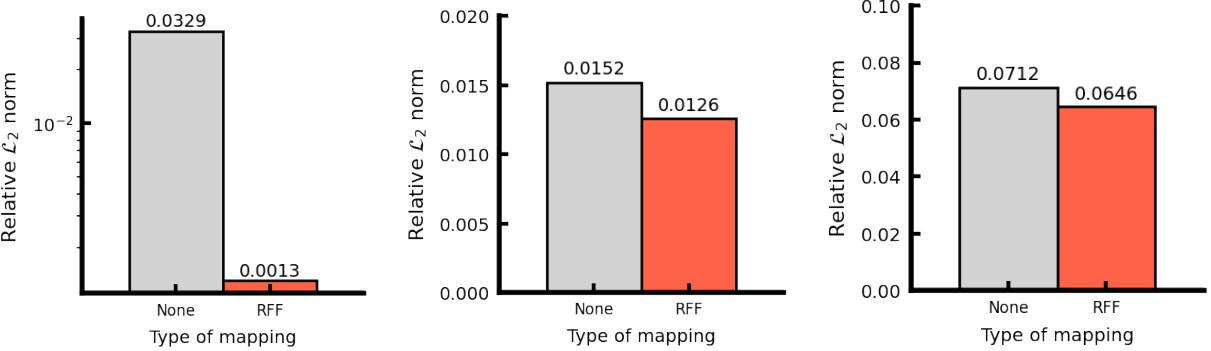

(a) Performance on Burgers' equation with 2048 resolution.

(b) Performance on Darcy flow with $141 \times 141$ resolution.

(c) Performance on NS2-full.

Figure 12: Ablation on the influence of RFF.

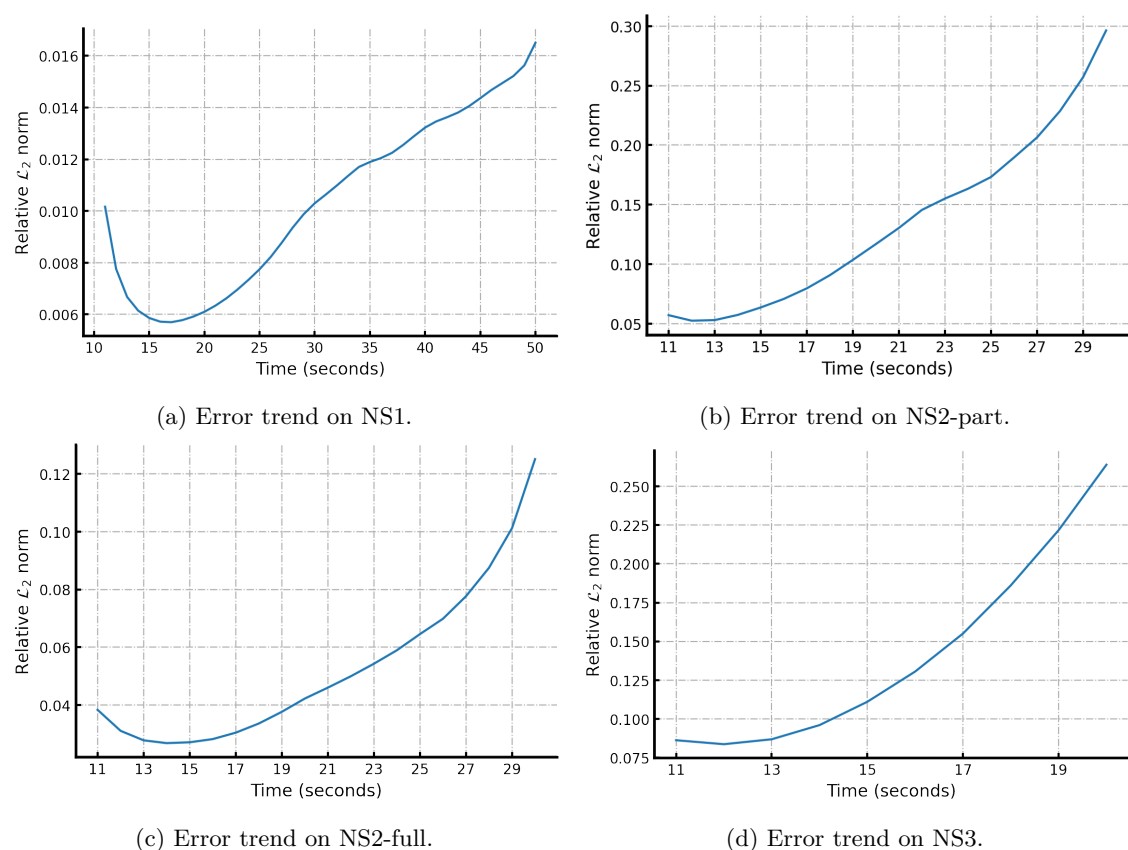

(a) Error trend on NS1.

(b) Error trend on NS2-part.

(c) Error trend on NS2-full.

(d) Error trend on NS3.

**Temporal trend of error**  We investigate the temporal trend of OFormer's prediction error on time-dependent system (Figure 13) by computing the mean error of model's prediction across all test sequences at a particular time step. In general, we observe that the error will accumulate exponentially as time grew. The compressible flow's error (Figure 13f) exhibits some levels of periodicity due to the intrinsic periodic property of the flow. Increasing the number of training samples alleviate the error, but does not change the

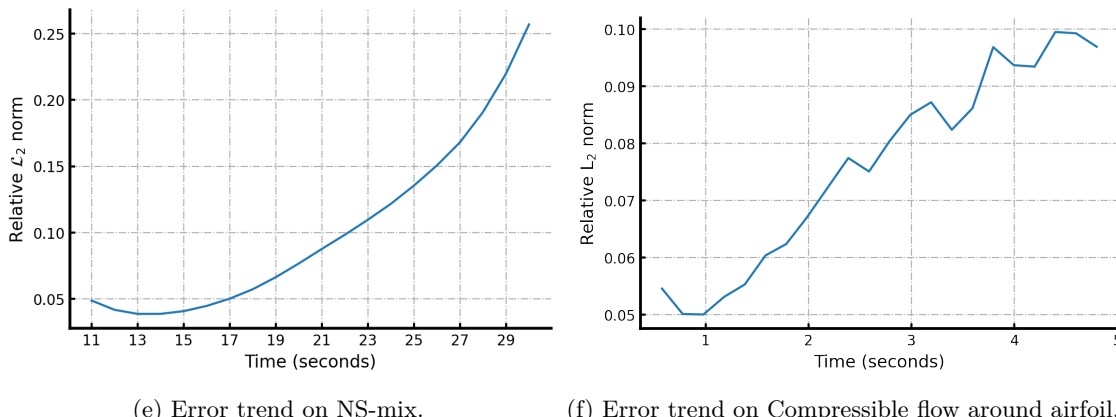

(e) Error trend on NS-mix.  (f) Error trend on Compressible flow around airfoil.

Figure 13: Error trend on different time-dependent systems.

exponential trend. How to alleviate the compounding prediction error for neural operator/PDE solver can be an interesting and important future venue.

## D   Dataset details

### D.1   Data generated on equi-spaced mesh

Following datasets[7] are used as benchmark in Li et al. (2021a); Cao (2021); Gupta et al. (2021), where the data is generated using equi-spaced simulation grid, and then downsampled to create dataset with different resolution.

**Burgers' Equation**   The nonlinear 1D Burgers' equation with the following form is considered:

$$\partial_t u(x,t) = -u\partial_x u(x,t) + \nu\partial_{xx} u(x,t), \quad x \in (0,1), t \in (0,1], \nu > 0,$$
$$u(x,0) = u_0(x), \tag{18}$$

where $\nu$ is viscosity coefficient, the equation is under periodic boundary conditions, and initial condition $u_0(x)$ sampled from Gaussian Random Field prior. The viscosity coefficient is set to 0.1. The learning objective is an operator mapping from initial condition $u_0(x)$ to the solution at time $t = 1$, i.e., $\mathcal{G} : u_0 \mapsto u(\cdot, 1)$. The data is generated via a split step method on 8192 resolution grid. At each time step, the state of the system is first advanced with the diffusion term and then the convection term (both calculated via Fourier transform), using the forward Euler scheme.

**Darcy Flow**   The steady-state 2D Darcy flow equation considered here takes the form:

$$-\nabla \cdot (a(x)\nabla u(x)) = f(x), \quad x \in (0,1)^2,$$
$$u_0(x) = 0, \quad x \in \partial(0,1)^2, \tag{19}$$

where $f(x)$ is the forcing function (set to constant 1) and we aim to learn the operator mapping diffusion coefficient $a(\cdot)$ to the solution $u(\cdot)$, i.e., $\mathcal{G} : a \mapsto u$. The coefficient function is sampled from a Gaussian Random Field prior with zero Neumann boundary conditions on the Laplacian kernel of the field. The data is generated via second-order finite difference solver on a $421 \times 421$ resolution grid.

---

[7]Dataset and numerical solver courtesy: `https://github.com/zongyi-li/fourier_neural_operator`

**Navier-Stokes Equation** The vorticity form of 2D Navier Stokes equation for the viscous, incompressible fluid reads as:

$$\partial_t \omega(x,t) + \mathbf{u}(x,t) \cdot \nabla \omega(x,t) = \nu \Delta \omega(x,t) + f(x), \quad x \in (0,1)^2, t \in (0,T],$$
$$\nabla \cdot \mathbf{u}(x,t) = 0, \quad x \in (0,1)^2, t \in (0,T], \tag{20}$$
$$\omega(x,0) = \omega_0(x), \quad x \in (0,1)^2,$$

where $\omega$ is vorticity, $\mathbf{u}$ represents velocity field, and $\nu$ is the viscosity, source term is set as: $f(x) = 0.1\left(\sin(2\pi(x_1 + x_2)) + \cos(2\pi(x_1 + x_2))\right)$, and initial condition $\omega_0(x)$ sampled from a Gaussian random field. The equation is under periodic boundary conditions. The goal is to learn an operator mapping the initial states of the vorticity field to the future time steps, i.e., $\mathcal{G} : \omega(\cdot,t)|_{t \in [0,10]} \mapsto \omega(\cdot,t)|_{t \in (10,T]}$. Generally, lower viscosity coefficients $\nu$ result in more chaotic dynamics and makes the learning more challenging.

The data is generated by solving the stream-function ($\psi : u = \nabla \times \psi$) formulation of above equation using pseudo-spectral method on a $256 \times 256$ grid. At each time step, the stream function is first calculated by solving a Poisson equation ($\nabla^2 \psi = -\omega$). Then the non-linear term is calculated by differentiating voriticity in the Fourier space, dealiased and then multiplied with velocities in the physical space. Lastly, the Crank-Nicolson scheme is applied to update the state.

## D.2 Data generated on non-equi-spaced mesh

**Poisson equation** The 2D Poisson equation for electrostatics and magnetostatics are defined as:

$$Electric: \quad -\nabla^2 U(x) = \frac{\rho(x)}{\epsilon(x)}, \quad x \in \Omega$$
$$Magnetic: \quad -\nabla^2 A_z(x) = \mu(x) I_z(x), \quad x \in \Omega \tag{21}$$

where $\rho$ is the charge density, $\epsilon$ is the permittivity of the material, $U$ is the electric potential, $\mu$ is the permeability of the material, $I_z$ is the z component of current density vector and $A_z$ is the z component of magnetic potential. The goal is to learn the solution operator of the Dirichlet boundary value problem: $\mathcal{G} : \{\rho, \epsilon\} \mapsto U \quad (U(x) = c \text{ for } x \in \partial\Omega)$ for electrostatics and $\mathcal{G} : \{I_z, \mu\} \mapsto A_z \quad (A_z(x) = c \text{ for } x \in \partial\Omega)$ for magnetostatics. In the problem we studied, $\epsilon(\cdot), \mu(\cdot)$ are held constant.

The data is generated by solving the above equation using finite element method implemented with FEniCS library (Alnæs et al., 2015) [8]. The element used is a quadratic triangle element. We use the pre-generated data from Lötzsch et al. (2022) for experiment.

**Euler equation of compressible flow** The Euler equation of compressible flow takes the form:

$$\partial_t \rho + \nabla \cdot (\rho \mathbf{u}) = f_1,$$
$$\partial_t(\rho \mathbf{u}) + \nabla \cdot (\rho \mathbf{u} \otimes \mathbf{u} + p\mathbb{I}) = \mathbf{f}_2,$$
$$\partial_t(\rho E) + \nabla \cdot (\rho E \mathbf{u} + p\mathbf{u}) = f_3, \tag{22}$$
$$\rho := \rho(x,t), \mathbf{u} := \mathbf{u}(x,t), p := p(x,t),$$
$$x \in \Omega, t \in [0,T],$$

where $\rho$ is the density, $\mathbf{u}$ is the velocity field, $p$ is the pressure, $E$ is the total energy per unit mass (which can be closed via $p = (\gamma - 1)\rho\left[E - 0.5(\mathbf{u} \cdot \mathbf{u})\right]$), and $f_1, \mathbf{f}_2, f_3$ are generic source terms. No-penetration condition is imposed at the airfoil. The solution operator of interest is: $\mathcal{G} : \{\mathbf{u}, \rho, p\}|_{x \in \Omega, t \in [0,0.576]} \mapsto \{\mathbf{u}, \rho, p\}|_{x \in \Omega, t \in (0.576,4.800]}$.

The dataset is generated by solving the above equation using finite volume method built in the SU2 library (Palacios et al., 2013). We use the pre-generated dataset[9] from Pfaff et al. (2020) to carry out the experiment. Note that different from Pfaff et al. (2020), we formulate this problem as a non-Markovian initial value problem and use a much large time step size ($\Delta t = 0.192$). In Pfaff et al. (2020), the learned solver is designed to learn the mapping: $u_t \mapsto u_{t+\Delta t}$ with $\Delta t = 0.008$ and then rollout the simulation with a Markovian setting.

---

[8]Dataset and numerical solver courtesy: `https://github.com/merantix-momentum/gnn-bvp-solver`
[9]Dataset courtesy: `https://github.com/deepmind/deepmind-research/tree/master/meshgraphnets`

# E    Further results and visualization

In this section we present the visualization of model's prediction on different problems. All error plots are based on point-wise absolute error.

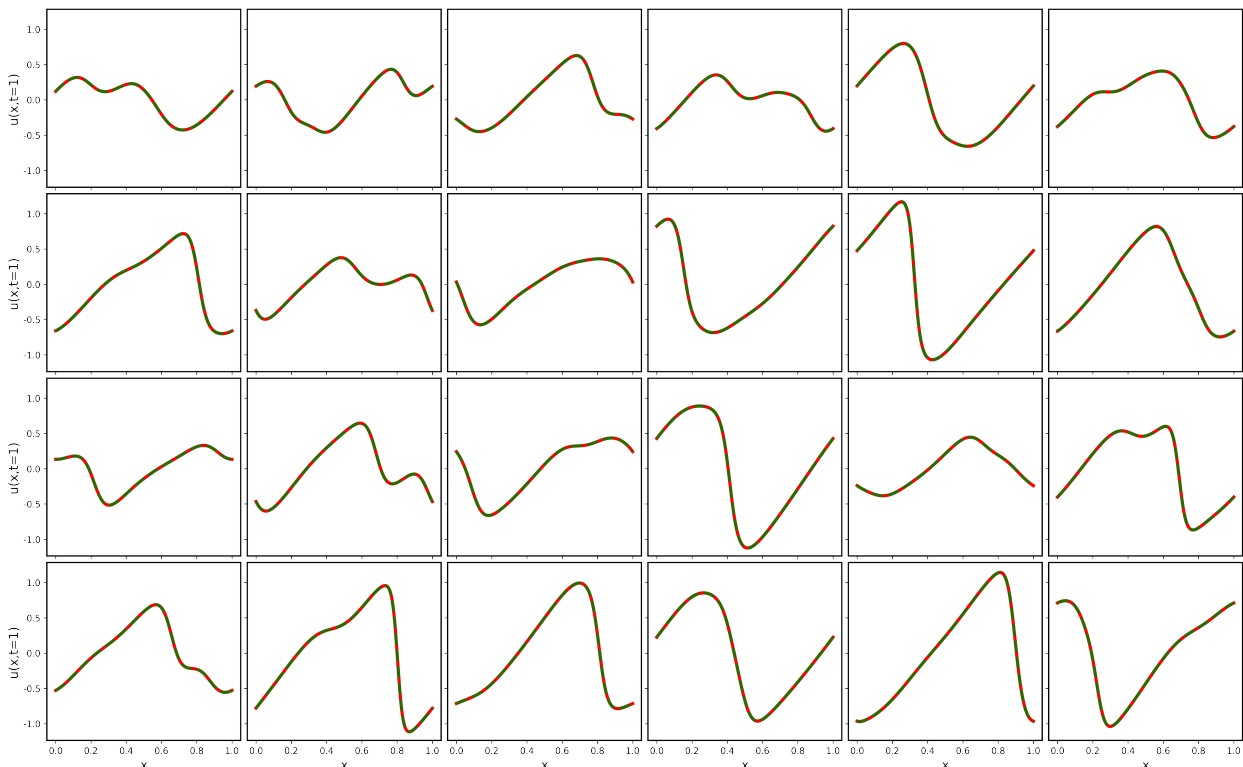

Figure 14: Model's prediction on 1D Burgers' equation, samples are randomly selected from test set. Green dotted lines denote the ground truth, and red lines denote model's prediction.

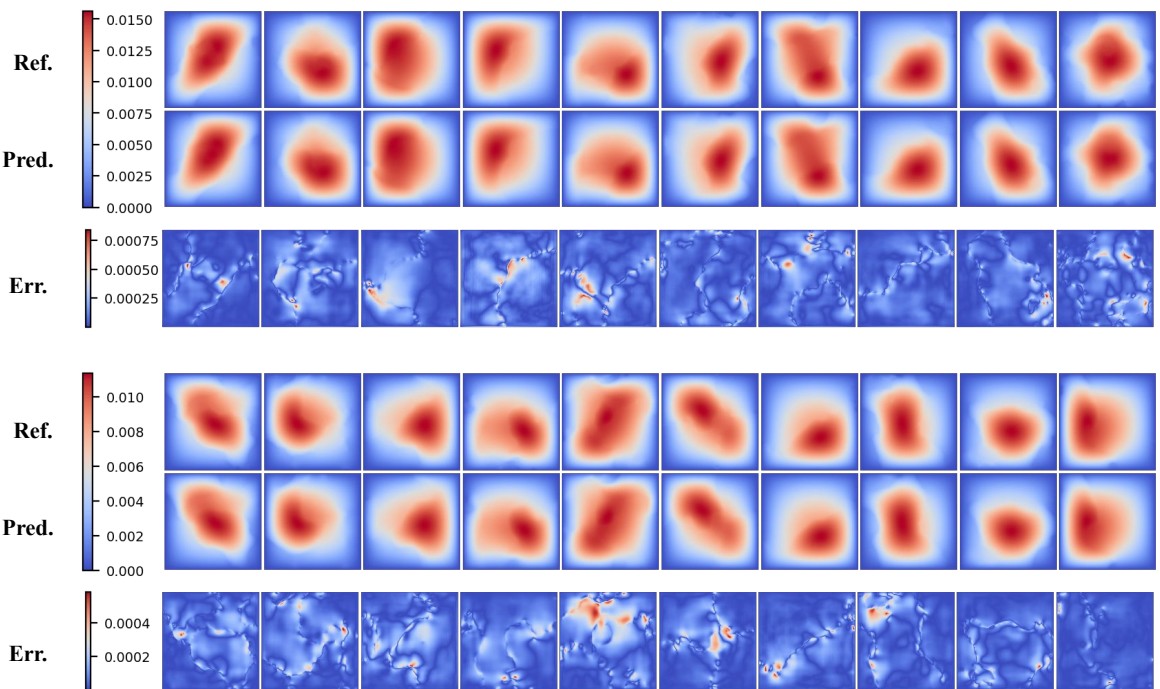

Figure 15: Model's prediction on 2D Darcy flow, samples are randomly selected from test set.

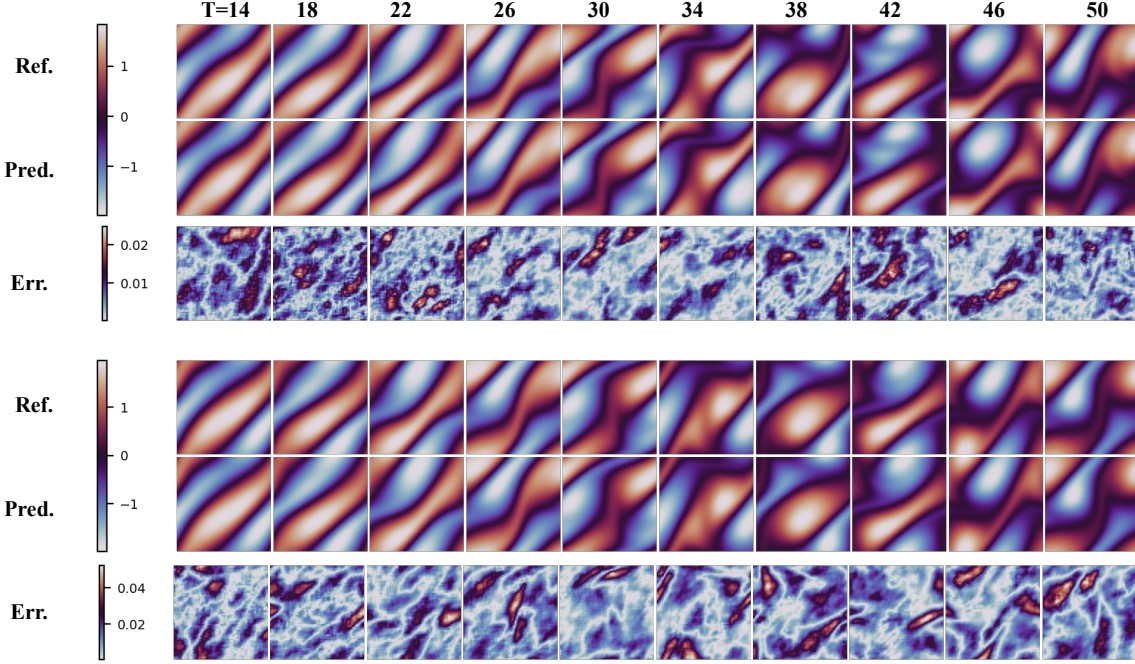

Figure 16: Model's prediction on NS1's test set ($\nu = 1e - 3$), with a Reynolds number around 20.

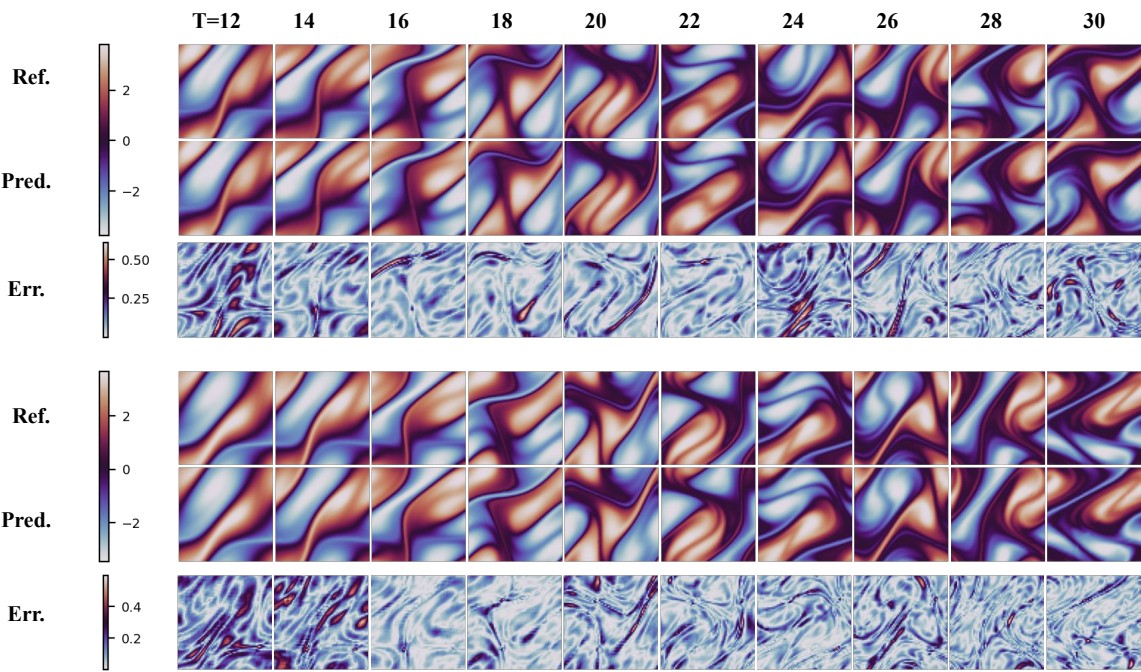

Figure 17: Model's prediction on NS2-full ($\nu = 1e-4$), with a Reynolds number around 200.

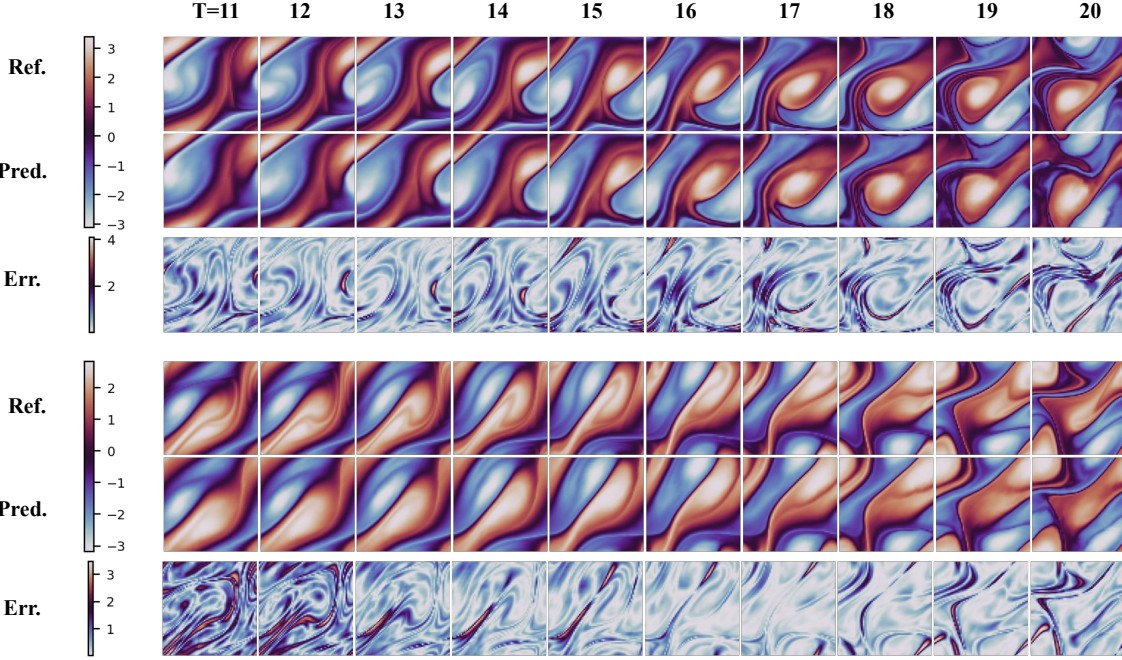

Figure 18: Model's prediction on NS3 ($\nu = 1e-5$), which has a Reynolds number around 2000.

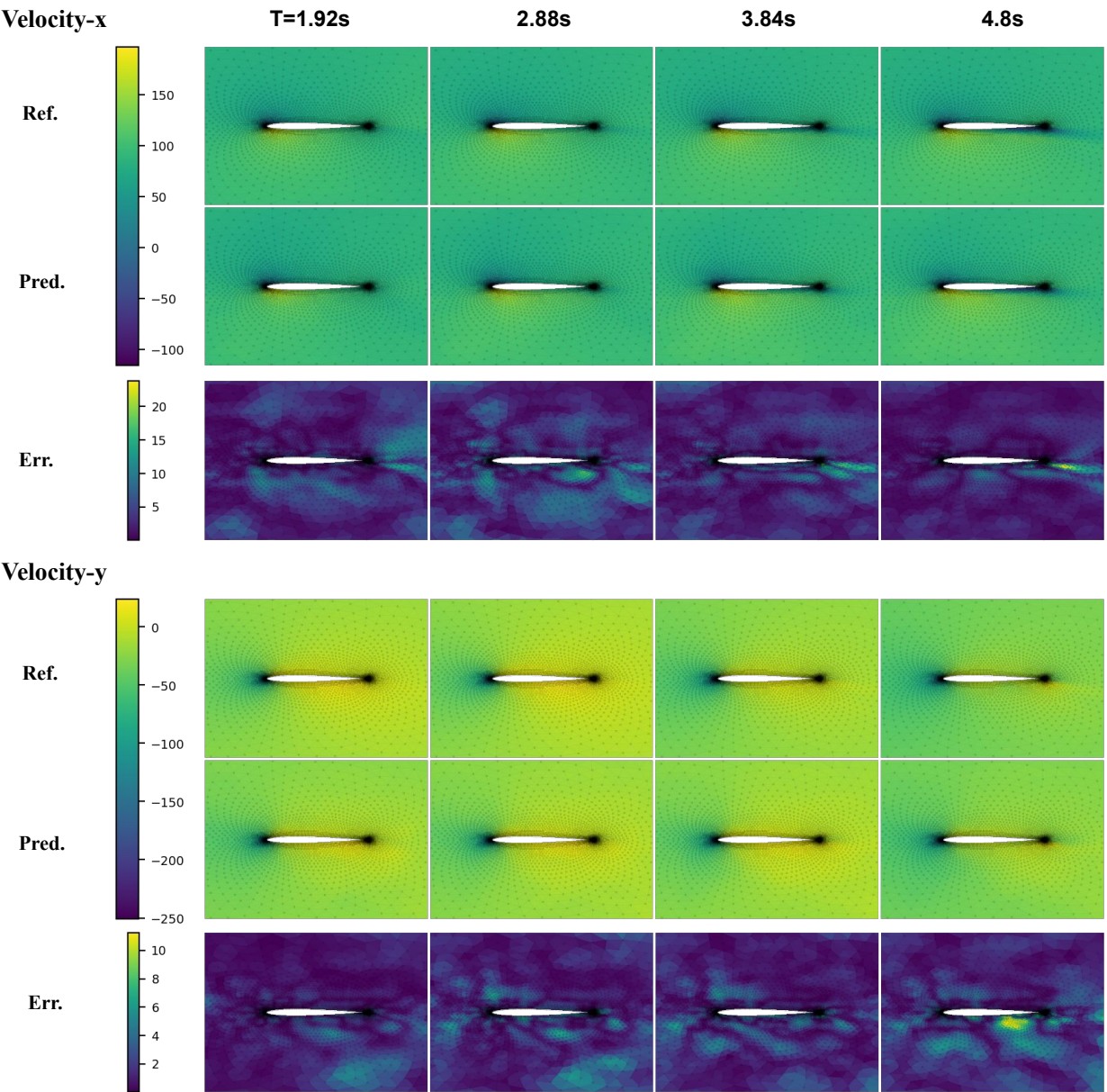

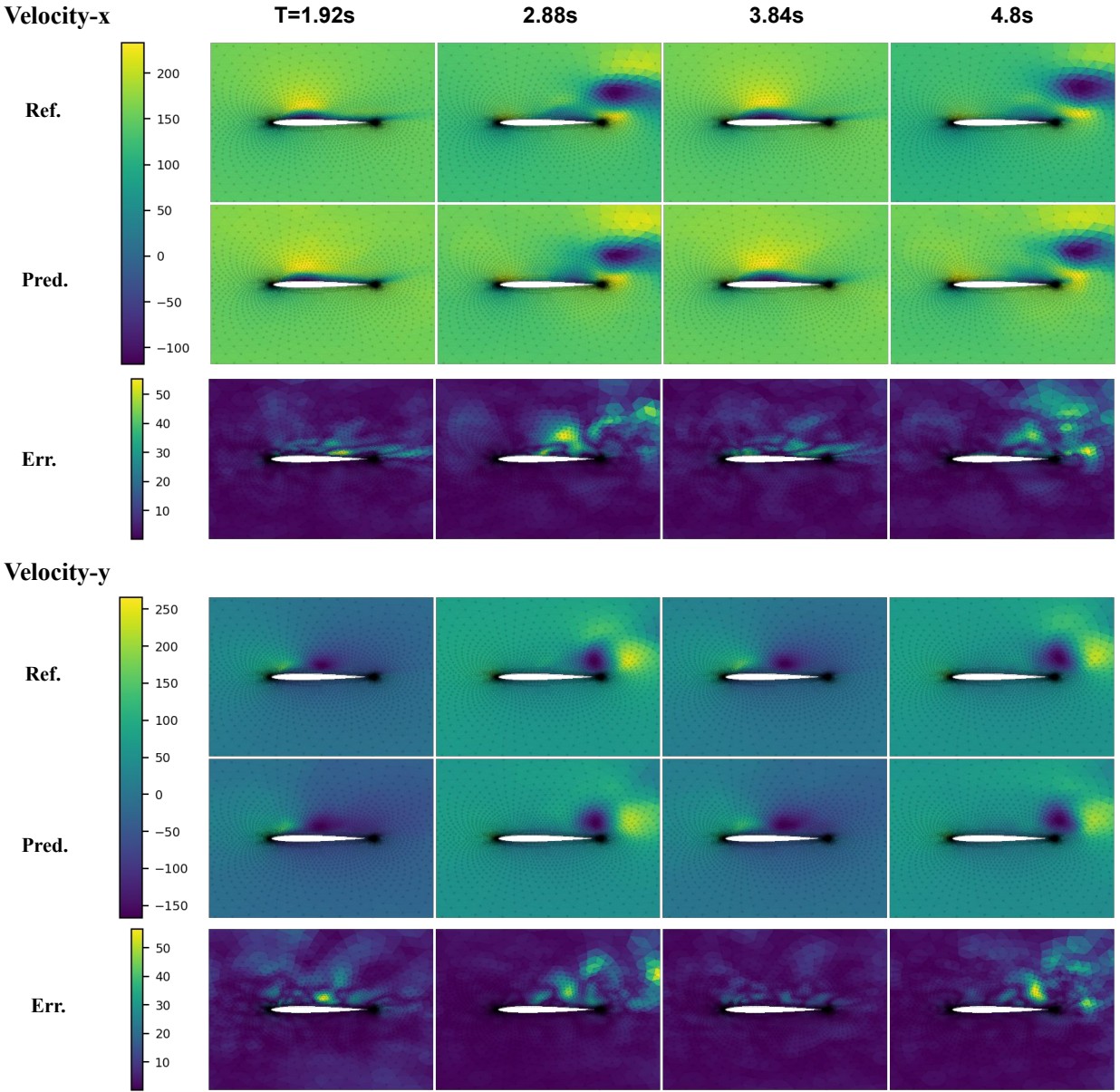

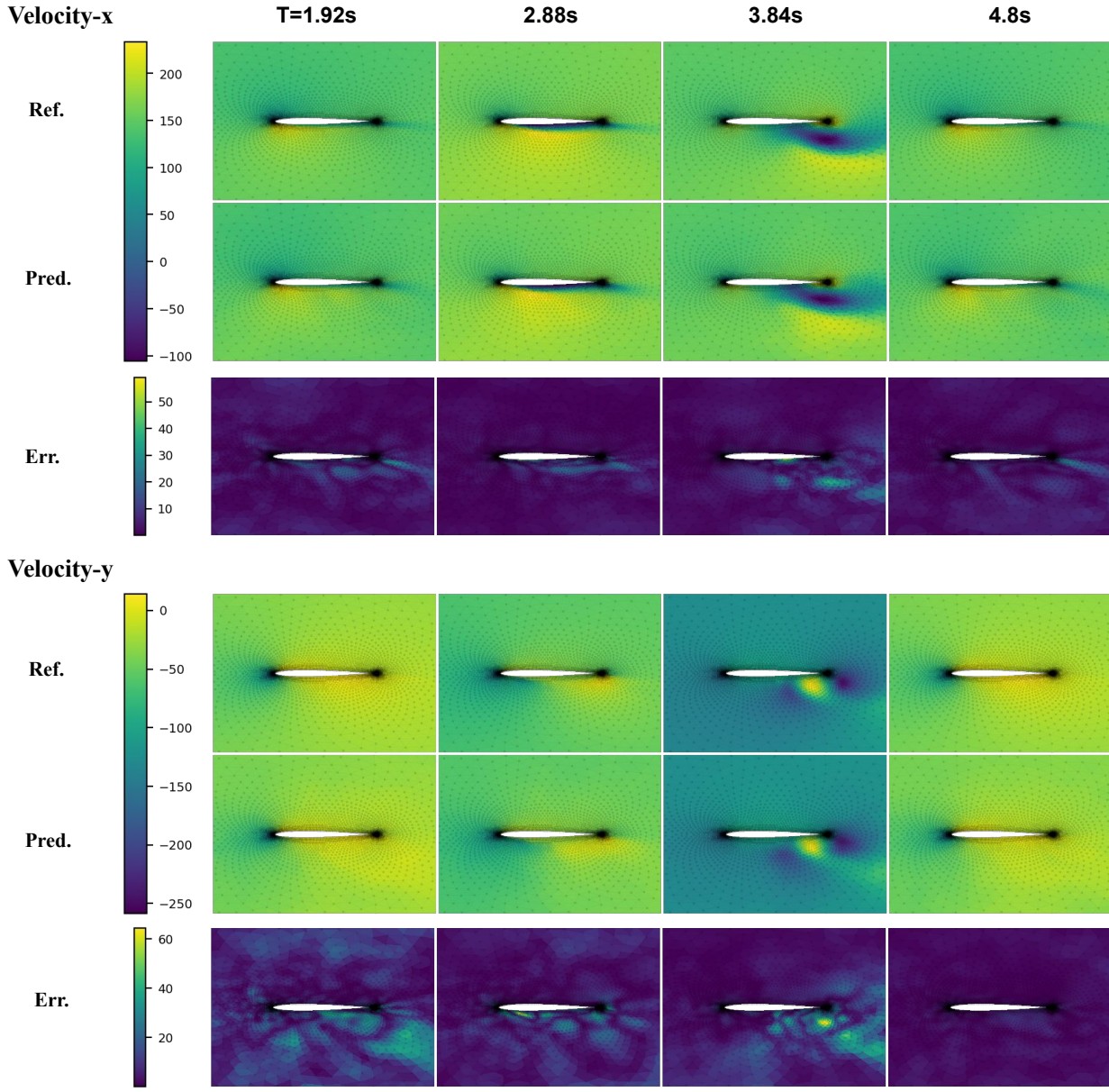

Figure 19: Model's prediction of the compressible flow around airfoil. The test root mean squared error of predicted momentum is **15.477** at plotted region (multiplying velocities with density on each node).

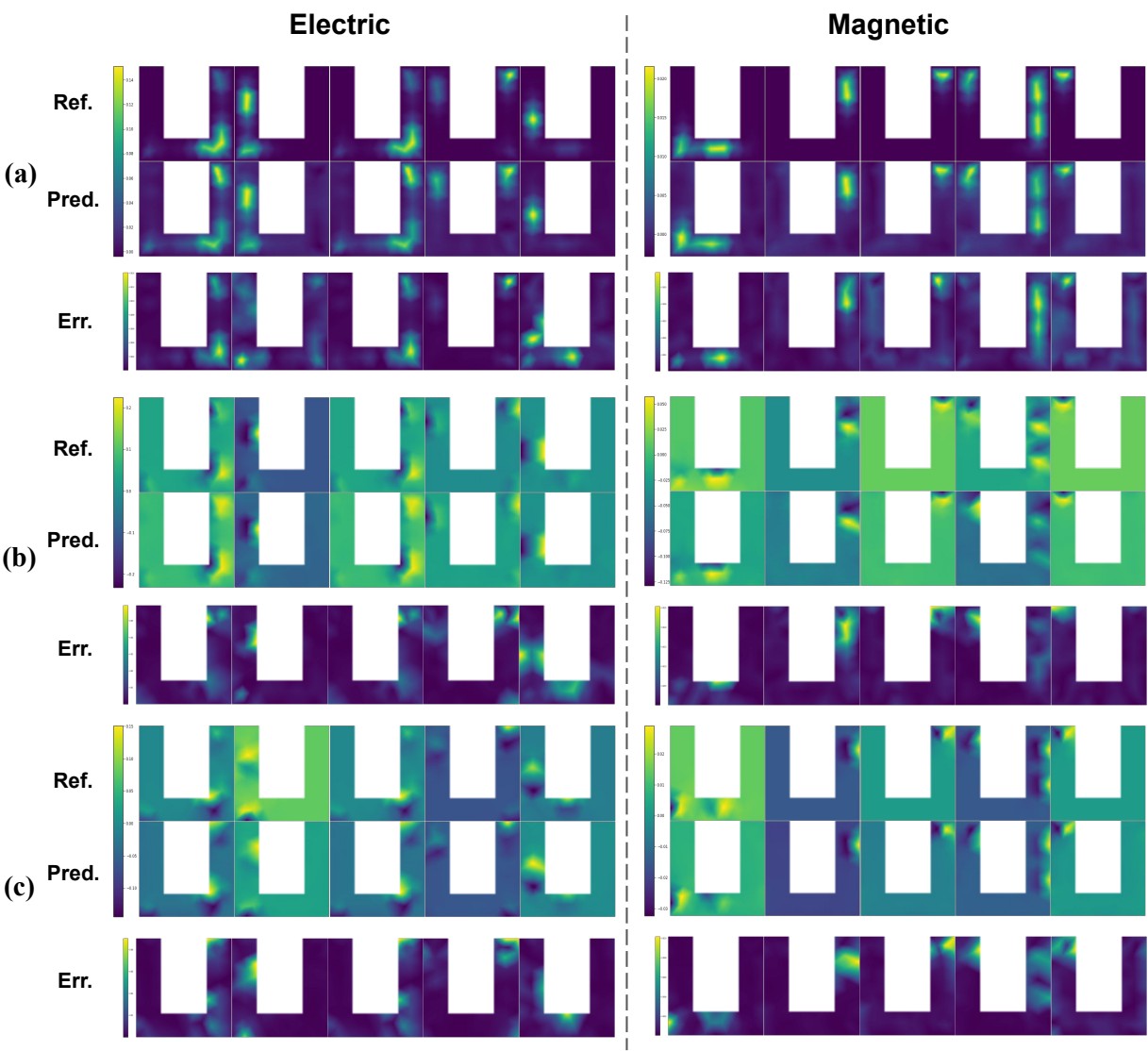

Figure 20: Model's prediction (interpolated from non-uniform grid points): **(a)** potential; **(b)** vector field's x component; **(c)**vector field's y component.

