# OpenReview forum: "Transformer for Partial Differential Equations’ Operator Learning"
_TMLR — Accepted by TMLR_

### Review · Reviewer_dyW3 · 2023-01-04

**Summary Of Contributions:**

The authors propose a new model to learn operators for PDEs from observed data. Their model, named OFormer, encodes the observed points and query points independently and then uses a cross-attention architecture to learn latent encodings. It then uses a recurrent architecture in the latent space to evolve these encodings forward in time. The authors show that their proposed model does not outperform the baselines on standard PDE benchmarks, but that it can lead to advantages when used on irregularly sampled grids, where the datasets are large and no mesh augmentation is used.

**Audience:**

Yes

**Broader Impact Concerns:**

No concerns.

**Claims And Evidence:**

Yes

**Requested Changes:**

As mentioned above, I would like to see error bars in the experimental results and at least one real-world dataset where the OFormer performs well.

**Strengths And Weaknesses:**

Strengths:
- Operator learning is an important problem
- The paper is easy to follow
- The model seems to perform well in the large-data, irregular-grid setting

Weaknesses:
- The model does not perform so well on the standard, regular-grid setting
- It is unclear how relevant the irregular-grid setting is in practice

Major comments:
- All the experimental results are lacking error bars, with makes it very hard to assess how significant the differences between the methods are. I would expect that the experiments would be run at least 3 times and for standard errors to be reported.
- Since the proposed OFormer only performs well in settings where the dataset is large, the grid is irregular and no mesh augmentation is used, it would be important to motivate when this setting is realistic. Are there any scenarios in practice where mesh augmentation could not be used? Similarly, are there real-world datasets where the grids are irregular and the query points are in different locations than the measured points? Maybe in weather/climate data or so? It would be good to demonstrate the performance of the OFormer on at least one such dataset.

Minor comments:
- Eq. (5): maybe use a different index than $i$ in the sum in the numerator, since it is easy to confuse with $i'$
- Eq. (8): should it say $z' = z^0 + \dots$?
- Sec. 4.1: imcompressible -> incompressible

---

> ### Author Response · Authors · 2023-01-23
> **Response to reviewer dyW3**
>
> We would like to thank the reviewer dyW3 for the constructive feedback. Here we address your concerns as below.
>
> ---
> > Comment:
> > * The authors show that their proposed model does not outperform the baselines on standard PDE benchmarks, but that it can lead to advantages when used on irregularly sampled grids, where the datasets are large and no mesh augmentation is used.
> > * The model does not perform so well on the standard, regular-grid setting.
>
> **Response**:
> We agree with reviewers that the advantage of the proposed model isn't very prominent on several regular grid benchmarks, especially those with stronger smoothing
> property like Burgers' and Darcy flow. However, we would also like to point out that on Navier-Stokes benchmark, the proposed model predicts the whole future sequence at once given the initial frames (by marching in the latent space).
> If we adopt a similar Markovian setting (i.e. given $t_n$ frame, the model predicts $t_{n+1}$ frame, then the $t_{n+1}$ frame is feed as input to the model) like FNO-2d and Galerkin Transformer, then we can observe clear improvement in low-data regime,
> which makes the model less data hungry. The l2 norm difference is shown as below.
>
> |Dataset |NS1|NS2-part|NS3|
> |-----|--------|---------------------|--------|
> |Default |0.0104 | 0.1708 | 0.1697 |
> |Markovian| 0.0088| 0.1475 | 0.1576 |
>
> ---
> > Comment:
> All the experimental results are lacking error bars, with makes it very hard to assess how significant the differences between the methods are.
> I would expect that the experiments would be run at least 3 times and for standard errors to be reported.
>
> **Response**:
> We appreciate the reviewer for the helpful suggestions. We updated the benchmark statistics of our models under three different random seeds in the revision version of manuscript.
>
> ---
> > Comment:
> > * Are there any scenarios in practice where mesh augmentation could not be used?
> > * Similarly, are there real-world datasets where the grids are irregular and the query points are in different locations than the measured points? Maybe in weather/climate data or so?
>
> **Response**:
> Note that mesh augmentation heavily relies on varying the distance between different nodes, this can be impractical for many PDE problems which are sensitive to the mesh (e.g. a high Reynolds number fluid simulation with solid geometries).
> In this case, it is much more difficult to choose an appropriate scale for changing the mesh density and shifting nodes.
>
> A wide array of physics simulation data are represented in irregular simulation meshes (e.g. a large class of elements in Finite Element Methods), or measurements from sparsely placed sensors. Although they can be regridded to uniform grids (through interpolation or statistical models like Gaussian
> process), this regridding process can be computationally expensive (as in some weather models) or compromises accuracy.
>
> As a proof-of-concept, we carried out an experiment on a climate prediction dataset - ERA5 reanalysis dataset from [1]. The reanalysis data is presented on a uniform grid by combining available observations and interpolation using statistical weather model.
> We evaluate the model based on both full-resolution input and sparse input (with 20% of randomly sampled observation points in full-resolution data). This mimics the scenario to predict only based on the sensory observations without running an expensive analysis model during the inference time.
> We report the latitude weighted RMSE as below:
>
> |Feature |3 days Geopotential|5 days Geopotential| 3 days Temperature | 5 days Temperature |
> |-----|--------|---------|-----------|-----------|
> |Full resolution |590|695|2.71 | 3.06|
> |Sparse| 660| 741 | 2.96 | 3.22 |
> |CNN baseline [1] |626 | 757|2.87 | 3.37|
>
> The exemplary visualization of geopotential's prediction can be viewed at this annonymized link: https://ibb.co/PZmWNpg
>
> [1] Stephan Rasp, Nils Thuerey, Data-driven medium-range weather prediction with a Resnet pretrained on climate simulations: A new model for WeatherBench, 2020
>
> ---
> > Minor comments:
> >  * Eq. (5): maybe use a different index than $i$ in the sum in the numerator, since it is easy to confuse with $i'$
> >  * Eq. (8): should it say $z'=z^0 + ...$?
> >  * Sec. 4.1: imcompressible -> incompressible
>
> **Response**:
> Thanks for the catch. We corrected them in the revision.

---

### Review · Reviewer_nrTx · 2023-01-12

**Summary Of Contributions:**

This work presents a deep learning model to solve partial differential equations (PDEs) using attention mechanisms. The model first encodes the input grid $x_1, \dots, x_n$ into a key $K$ and value $V$ matrices using self-attention, and then encodes a different evaluation grid $y_1, \dots, y_m$ into a query $Q$ matrix. The model then uses cross-attention to generate a latent representation for each instance $y_i$, which is then fed through an MLP $T$ times to simulate $T$ time steps of the equation. Finally, another MLP generate the prediction for the input $y_i$.

The main advantange of this work over others lays in the use of attention to encode the input, which allows the model to have different discretizations for the input and output grids, as well as using irregular grids. The latter could be achieved before with GNNs, yet the authors show that their architecture (OFormer) outperforms a proposed GNN architecture when no mesh-augmentation is applied.

**Audience:**

No

**Broader Impact Concerns:**

None.

**Claims And Evidence:**

Yes

**Requested Changes:**

**Questions:**
- Q1. Why not using relative encoding for the input grid ($x$) too?

**Required changes**
1. Clarity should be improved. The model should be well described (e.g., explain what is the output of each module, add notation to the figures, cross-attention should be described somewhere, etc.).
2. If one of the advantages is the ability to use two grids, this should be showed/exploited in some way during experimentation.
3. A clear motivation/advantage of using the proposed model over existing ones should still be found. While I know this is non-actionable, I can point out to some directions that come to mind.
- Equation (4) shows that $K^\top V$ could be pre-computed. Could you pre-compute it using the training data, and drop the first part of the encoder during testing?
- Following on the previous point, what if the first part of the model contained information about the target, such that you could exploit training information?

**Desirable changes**

4. The connections with Transformer should be made clearer. To my understanding, _the entire encoder_ is a Transformer as in the original paper, with the only difference being that there is only positional encodings as inputs. This interpretation would ease a lot the encoder description.

**Strengths And Weaknesses:**

**Strengths**
- S1. The use of attention to solve PDEs is interesting, specially, taking advantage of the work by Cao to use attention without softmax to save computations.
- S2. The model enables independent input-output meshes, as well as irregular grids.
- S3. The proposed model has similar performance to existing methods.
- S4. The experimental section is well done and honest, having an ablation study as well as a small qualitative study of the latent space.

**Weaknessess**
- W1. Writting could be improved. For example, the architecture description is not super clear. It took me several reads and looks at the code to understand how all the different parts of the model interconnect.
- W2. Having a different input and output grid seems really odd/niche to me. Indeed, to my understanding this is not utilized at all in the experiments.
- W3. While not seeking sota, the scenarios to prefer the proposed model over others are scarce (irregular grids where mesh-augmentation is not an option).
- W4. To my understanding the main contribution is the usage of a Transformer (as latent marching has been used in the past), making the contributions rather weak.

---

> ### Author Response · Authors · 2023-01-23
> **Response to reviewer nrTx - 1**
>
> We would like to thank reviewer nrTx for the insightful comments. Here we address your concerns as below.
>
> ---
> > Comment:
> > * Writting could be improved. For example, the architecture description is not super clear. It took me several reads and looks at the code to understand how all the different parts of the model interconnect.
> > * Clarity should be improved. The model should be well described (e.g., explain what is the output of each module, add notation to the figures, cross-attention should be described somewhere, etc.).
>
> **Response**:
> We updated the schematic (Figure 1) to clarify where RoPE is used and the usage of Cartesian coordinates. Below we also provide a brief description of each component used in the model to clarify their roles:
>
> * Encoder: The encoder takes the sampling of the input function value $u(x_1), ..., u(x_n)$ as input (the input function can be the initial condition of the system or some coefficient function). The Cartesian coordinates of the
> points where these function values are sampled ( $x_1, ..., x_n$ ) are also inputted as the positional encoding. So the input are $u(x_i)$ and $x_i$. The input are then processed by several self-attention blocks. Note that the Cartesian coordinates
> of the input points are also used to produce relative positional encoding in every self-attention block.
>
> * Decoder: The decoder comprises a coordinate-based network, a propgator (for time-dependent system) and a decoding network.
> The coordinate-based network takes the Cartersian coordinate of query points as input, and the propagator propagate the dynamics in the latent space.
>
> > Comment:
> > Q1. Why not using relative encoding for the input grid $x_1, ...., x_n$ too?
>
> **Response**:
> Following last response, we do use relative encoding for the encoder.
>
> ---
> > Comment:
> > * Having a different input and output grid seems really odd/niche to me. Indeed, to my understanding this is not utilized at all in the experiments.
> > * If one of the advantages is the ability to use two grids, this should be showed/exploited in some way during experimentation.
>
> **Response**:
>
> In the original manuscript, we have provided an example of applying model to different input/output grid at Page 10, an exemplary visualization at Figure 5, and a quantitative benchmark (Table 13,14) in the Section C of Appendix.
> Based on the reviewer's feedback, we rearranged the paragraph "Application to irregular grids" to make these results clearer in the revision.
>
> The ability to query at any arbitrary location makes the model better fit in with the concept of Operator learning as proposed in DeepONet [1] and Neural Operator [2], where the output of model is a function
> instead of a scalar or a fixed length vector. From a practical perspective, such property enables using model for sparse reconstruction, super-resolution and makes it generalizeble to different discretization in inference time.
> A related class of deep learning model that enjoys similar properties is Implicit Neural Representation (INR), where it can be used to do image super-resolution [3] or novel view synthesis [4].
>
> [1] Lu, Lu, Pengzhan Jin, and George Em Karniadakis. "Deeponet: Learning nonlinear operators for identifying differential equations based on the universal approximation theorem of operators." arXiv preprint arXiv:1910.03193 (2019).  \
> [2] Kovachki, Nikola, et al. "Neural operator: Learning maps between function spaces." arXiv preprint arXiv:2108.08481 (2021).  \
> [3] Chen, Yinbo, Sifei Liu, and Xiaolong Wang. "Learning continuous image representation with local implicit image function." Proceedings of the IEEE/CVF conference on computer vision and pattern recognition. 2021.  \
> [4] Mildenhall, Ben, et al. "Nerf: Representing scenes as neural radiance fields for view synthesis." Communications of the ACM 65.1 (2021): 99-106.
>
> ---
> > Comment:
> > While not seeking sota, the scenarios to prefer the proposed model over others are scarce (irregular grids where mesh-augmentation is not an option).
>
> **Response**:
>
> As mesh-augmentation involves varying the distance between nodes in the mesh, it could be difficult to choose an appropriate augmentation scheme for a lot of mesh sensitive problem (e.g. an advection dominated system as in many fluid related problems) and thus making it less applicable.
> In addition, as shown in Table 1, we also observe the model shows relatively strong performance on time-dependent problem with sufficient data.
>
> (Continued next reply)

---

> > ### Author Response · Authors · 2023-01-23
> > **Response to reviewer nrTx - 2**
> >
> > ---
> > > Comment:
> > > * To my understanding the main contribution is the usage of a Transformer (as latent marching has been used in the past), making the contributions rather weak.
> > > * The connections with Transformer should be made clearer. To my understanding, the entire encoder is a Transformer as in the original paper, with the only difference being that there is only positional encodings as inputs. This interpretation would ease a lot the encoder description.
> >
> > **Response**:
> >
> > We updated the manuscript to include more description about the connection of the proposed model and original Transformer (Section 3.2, page 4). The proposed model is similar to the original Transformer in principle. Practically, the major differences in the proposed model are: (1) Use relative positional encoding; (2) Remove Softmax and use linear variant; (3) Use instance normalization for Q/K (or K/V in 1D problem); (4) Use a recurrent neural network architecture in the decoder side instead of temporal attention. In addition to positional encodings, the encoder also takes input function values as input.
> >
> > We agree with reviewer that latent marching has been used in the past (as we have also mentioned this in related works), but we would also like to point out several differences compared to the existing models. First, most of previous works rely on a convolutional autoencoder architecture to encode/decode the state of the system, which makes the latent/output tied to a specific grid and resolution.
> > The proposed model does not suffer from this issue. In the proposed model, we use a coordinate-based network for initializing latent encoding (on each query point), and then the input information is aggregated to these query points via cross-attention with encoding output by the encoder. This process is agnostic to mesh topology and can query at arbitrary locations. The proposed point-wise latent marching architecture is simple and efficient since it does not require spatial interaction between different query points during dynamics propagation. Furthermore, it can be viewed as a latent ODE (Ordinary Differential Equation) with fixed time interval,
> > which can naturally be extended into a continuous model by replacing the MLP with a neural ODE [1] and allows model to be even temporally continuous (i.e. can query at arbitrary time steps).
> >
> > [1] Chen, Ricky TQ, et al. "Neural ordinary differential equations." Advances in neural information processing systems 31 (2018).
> >
> > ---
> > > Comment:
> > > A clear motivation/advantage of using the proposed model over existing ones should still be found. While I know this is non-actionable, I can point out to some directions that come to mind.
> > > * Equation (4) shows that  could be pre-computed. Could you pre-compute it using the training data, and drop the first part of the encoder during testing?
> > > * Following on the previous point, what if the first part of the model contained information about the target, such that you could exploit training information?
> >
> > **Response**:
> >
> > Since the input of the encoder is input function values $u(x_1), ..., u(x_n)$ and its corresponding sampling grid $x_1, ..., x_n$ , which varies across different samples and therefore cannot be pre-computed. However, if the query grid stays the same, the $Q$ part in the cross-attention can indeed be pre-computed.
> > And because the input function values and target function values vary across samples, training data does not contain information about the target function values in validation data.

---

> > > ### Comment · Reviewer_nrTx · 2023-02-27
> > > **Thanks for the clarifications**
> > >
> > > Dear authors, thank you for the changes and for the detailed response. Most (if not all) of my concerns have been addressed.

---

### Review · Reviewer_duoN · 2023-02-11

**Summary Of Contributions:**

This paper further advances the methodology of PDE operator learning using Transformers proposed in (Cao, 2021). The main contribution is the adaptation of several techniques popularized by the Transformer research in NLP/CV, such as cross attention and relative positional encodings, to the context of PDE operator learning to make inference on non-equispaced grid possible. Overall it is a worthy addition to the literature if the writings and presentation can be improved (see below).

**Audience:**

Yes

**Claims And Evidence:**

Yes

**Requested Changes:**

- add a summary of the contribution after the introduction or literature review.
- The number of parameters in DeepONets depends on the number of query points (grid size), since the final summation is with respect to the number of query points. While for Transformer there is no such dependence, the final sum is with respect to the embedding dimension $d$ that has nothing to do with the discretization size. It would be nice to add this clarification.
- add an ablation study of the first layer (random Fourier projection vs vanilla MLP).
- some clarification needs to be added for RoPE vs Euclidean coordinates. In (7) it is said that the random Fourier projection with Euclidean coordinates is used as the first layer, later on page 7 the Rotary Positional Embedding is used. As positional encodings are simply coordinates of discretizations in operator learning, if Euclidean coordinates are used in the first layer, where exactly is RoPE used? It would be better to add it on the diagram.
- a table or a plot of numerical values of relative errors at different time steps should be given, instead of heat maps.

### Things that need to be clarified
- Page 2: "an general architecture".
- Page 3: "$d$ being the dimension of vectors", here $d$ should be embedding/latent dimension or channels.
- Page 3: $\mathbf{q}_i\cdot \mathbf{k}_s$ should be approximating the kernel evaluated at $k(x_i, x_s)$.
- Page 4: the writing on mathematics can be improved. For example, "$q_l(y_{i'})$ as the trunk net" is inaccurate as $q_l(y_{i'})$ is a scalar while a neural net is a mapping.
- Page 4: is it LayerNorm or instance normalization used in the model?
- Page 5, equation (8): in the first equation, $\mathbf{z}'$ is used before being defined.
- Page 8, magnetic and electric potentials are different, the current notation is confusing as the same $u$ is used. Moreover, in 2D, it is better to use the $\text{rot}$ operator instead of $\nabla \times$. For example in Girault-Raviart, $\nabla^\perp$ is used in 2D.
- Page 24: suggestion on expressing operator maps: it is better to use $\mathcal{G}: a\mapsto u$ directly instead of $\mathcal{G}: a(x)\mapsto u(x)$ as $a$ denotes a function while $a(x)$ means sampling $a$ at $x$.
- Page 25: The FEniCS's reference is not quite appropriate (currently given as (Scroggs et al., 2022)). "The element used is a 2-degree triangle element". I am not sure what "2-degree" is referring here.


**Strengths And Weaknesses:**

### Strengths
- The connection of the cross attention with DeepONets is interesting.
- Novel treatment for spatial-temporal problems using a shared propagator.
- Comprehensive numerical experiments.

### Weaknesses
- The writings could enjoy some further polishing; see also request changes below for some suggestions.
- No comparison with DeepONets in the non-equispaced case.
- The usage of Rotary Position Embedding is not supported by a specifically designed experiment.
- The modifications to the existing architectures lack theoretical motivations.

---

> ### Author Response · Authors · 2023-02-23
> **Response to duoN - 1**
>
> We would like to thank the reviewer duoN for the insightful comments on improving the paper. Here we address your concerns and provide some clarification as below.
>
> ---
> > Motivation of using Rotary positional Encoding (RoPE).
>
> **Response**:
> The use of RoPE is motivated by the (non-linear) learnable integral kernel: $\int\kappa_{\theta}(x,y, v(x), v(y))v(y)dy$ which is proposed in [1] (described as "Fourier type" attention in [2]).
> Prior works have explored paramterizing $\kappa_{\theta}(x,y, v(x), v(y))$ using message-passing neural network [3] or using a convolution kernel $\kappa_{\theta}(x-y)$ as in [4].
>
> Our proposed attention layer with RoPE can be viewed as parametrizing the kernel using:
> $$\kappa_{\theta}(x,y, v(x), v(y))=[W_qv(x)]^T\Theta(x-y)W_kv(y),$$ where $v(x), v(y)\in \mathbb{R}^{c}$ is the input function, $W_q, W_k \in \mathbb{R}^{d \times c}$ are learnable affine transformations
> and $\Theta(\cdot)$ are the RoPE (defined in equation (9) in the main manuscript). The integral kernel here does not explicitly depend on $(x, y)$, instead it depends on the relative distance $x-y$ and $(v(x), v(y))$.
>
> [1] Li, Zongyi, et al. "Neural operator: Graph kernel network for partial differential equations." arXiv preprint arXiv:2003.03485 (2020).
>
> [2] Cao S. Choose a transformer: Fourier or galerkin[J]. Advances in neural information processing systems, 2021, 34: 24924-24940.
>
> [3] Li, Zongyi, et al. "Multipole graph neural operator for parametric partial differential equations." Advances in Neural Information Processing Systems 33 (2020): 6755-6766.
>
> [4] Li, Zongyi, et al. "Fourier Neural Operator for Parametric Partial Differential Equations." International Conference on Learning Representations.
>
> ---
> > No comparison with DeepONets in the non-equispaced case.
>
> **Response**:
> Many of the non-equispaced cases we studied involve varying input grids and includes different testing geometries, while vanilla DeepONet can only be trained on a grid at one time. Therefore we opt for not re-implementing and reporting DeepONets on them to avoid misleading conclusion.
>
> ---
> > Comment: add a summary of the contribution after the introduction or literature review
>
> **Response**:
> We added a summary of contribution in the last paragraph of the introduction.
>
> ---
> > Comment: The number of parameters in DeepONets depends on the number of query points (grid size), since the final summation is with respect to the number of query points.
> >  While for Transformer there is no such dependence, the final sum is with respect to the embedding dimension  that has nothing to do with the discretization size. It would be nice to add this clarification.
>
> **Response**:
> We appreciate the reviewer for pointing this out. We added a description of this property in the "Cross-attention" paragraph on page 4.
>
> ---
> > Comment: add an ablation study of the first layer (random Fourier projection vs vanilla MLP).
>
> **Response**:
> We added an ablation study of RFF (random Fourier features) vs vanilla MLP to the Appendix Section C, page 24-25.
>
> ---
> > Comment: some clarification needs to be added for RoPE vs Euclidean coordinates. In (7) it is said that the random Fourier projection with Euclidean coordinates is used as the first layer, later on page 7 the Rotary Positional Embedding is used. As positional encodings are simply coordinates of discretizations in operator learning, if Euclidean coordinates are used in the first layer, where exactly is RoPE used?
> > It would be better to add it on the diagram.
>
> **Response**:
> Based on reviewer's suggestion, we updated the schematic (Figure 1) to clearly mark where RoPE is used.
> To clarify, it is used in both the self-attention and cross-attention layer.
> More specifically, before cross-attention, the Euclidean coordinate $y_i$ of query point is first sent to random Fourier projection $\gamma$ and then a MLP $\phi$:
> $$q_i=\phi(\gamma(y_i))$$
>
> The $q_i$ is then modulated by RoPE: $q'_i=\Theta(y_i)q_i$ and sent to cross-attention. In practice we find that using RoPE in cross-attention also benefits the final performance.
>
> ---
> > Comment: a table or a plot of numerical values of relative errors at different time steps should be given, instead of heat maps.
>
> **Response**:
> We added the plots of error vs time to the Appendix Section C, page 25-26.
>
> (Continued next reply)

---

> > ### Author Response · Authors · 2023-02-23
> > **Response to duoN - 2**
> >
> > ---
> > > Page 2: "an general architecture".
> >
> > **Response**:
> > The original description is vague, we have updated it to "architecture based on universal operator approximation theorem".
> >
> > ---
> > > Page 3: " being the dimension of vectors", here  should be embedding/latent dimension or channels.
> >
> > **Response**:
> > We updated the description from "dimension" to "channel".
> >
> > ---
> > > Page 3: $\mathbf{q}_i\cdot \mathbf{k}_s$ should be approximating the kernel evaluated at $k(x_i, x_s)$
> >
> > **Response**:
> > We updated the description as suggested.
> >
> > ---
> > > Page 4:  the writing on mathematics can be improved. For example, " $q_l(y_{i'})$ as the trunk net" is inaccurate as $q_l(y_{i'})$ is a scalar while a neural net is a mapping
> >
> > **Response**:
> > We updated this sentence to " $q_l(y_{i'})$ as the output of the trunk net" (and similarly for branch net).
> >
> > ---
> > > Page 4: is it LayerNorm or instance normalization used in the model?
> >
> > **Response**:
> > We use instance norm inside attention, i.e. after applying projection matrix $W_q, W_k, W_v$ to the input.
> > Layer norm is used between different blocks, for instance, after every feed forward network.
> >
> > ---
> > > Page 5, equation (8): in the first equation, $z'$ is used before being defined.
> >
> > **Response**:
> > We corrected the spotted typo, it should be $z^{(0)}$.
> >
> > ---
> > > Page 8, magnetic and electric potentials are different, the current notation is confusing as the same $u$ is used.
> > Moreover, in 2D, it is better to use the $\text{rot}$ operator instead of $\nabla \times$. For example in Girault-Raviart, $\nabla^\perp$ is used in 2D.
> >
> > **Response**:
> > We updated the description regarding magnetic potential and eletric potential. As suggested, we use $\mathbf{E}$ to denote the electric field, $\mathbf{B}$ to denote the magnetic field, and $\text{rot}$ to denote the curl operator.
> >
> > ---
> > > Page 24: suggestion on expressing operator maps: it is better to use $\mathcal{G}: a\mapsto u$ instead of $\mathcal{G}: a(x)\mapsto u(x)$.
> >
> > **Response**:
> > We corrected the expression based on reviewer's suggestion.
> >
> > ---
> > > Page 25: The FEniCS's reference is not quite appropriate (currently given as (Scroggs et al., 2022)). "The element used is a 2-degree triangle element".
> > I am not sure what "2-degree" is referring here.
> >
> > **Response**:
> > We changed the reference of FEniCS to Alnæs et al., 2015. We also updated the ambiguous word "2-degree" to "quadratic".

---

> > > ### Comment · Reviewer_duoN · 2023-02-27
> > > **Comments on the revision**
> > >
> > > It is nice that the authors added so many things in such a short response window. I would hold my original judgement "it is a worthy addition to the literature" without the condition mentioned earlier in the review.

---

### Author Response · Authors · 2023-02-23
**General response**

We want to thank all the reviewers for their efforts and the insightful comments on our work.

Below we summarize the main modification made to the manuscript. We have also addressed detailed comments of every reviewer in separate replies.
Please don’t hesitate to let us know if there is any further question or additional comment.

* Updated Figure 1 to mark where RoPE is used.
* Added a summary of contribution in the last paragraph of Introduction section.
* Added a discussion on the connection between proposed architecture and original Transformer (page 4, Section 3.2 Model Architecture).
* Added error bars for OFormer's benchmark result (Table 1/2/3/4).
* Added an ablation study on random Fourier features (RFF) vs MLP (page 24- 25, Appendix Section C).
* Added plots on temporal error evolution trend of time-dependent systems (page 25-26, Appendix Section C).

We have also updated the writing of some sentences based on reviewers' suggestions and corrected spotted typos.

---

### Decision · Action_Editors · 2023-04-17

**Recommendation:** Accept as is

**Comment:**

The authors have successfully addressed all the review comments, and all the reviewers agree that the paper now meets the bar, and the audience of TMLR will be interested in reading it.

**Audience:**

Yes

**Claims And Evidence:**

Yes